# PonderLM-2: Pretraining LLM with Latent Thoughts in Continuous Space

**Boyi Zeng** [1]  **He Li** [1]  **Shixiang Song** [1 3]  **Yixuan Wang** [1 3]  **Zitong Wang** [1 5]  **Ziwei He** [3]  **Xinbing Wang** [4]
**Zhouhan Lin** [1 2]

## Abstract

The remarkable success of Chain-of-Thought (CoT), which enhances performance by scaling generation steps at test-time, inspires us to ask: can we leverage a similar scaling of computational steps during pretraining to improve the generation of each individual token? To address this, we propose a novel pre-training methodology: *Pretraining Language Models with Latent Thoughts (PonderLM-2)*. Our approach pretrains a language model (LM) to first generate an intermediate latent thought—the last hidden state of the current position—which is then used as input to predict the actual subsequent token. This additional computational step enables the LM to refine its prediction within unconstrained continuous space. Our experiments demonstrate that, at an identical inference cost, a LM that generates one additional latent thought per token outperforms a standard model with *double* the parameters. For instance, our PonderLM-2-Pythia-1.4B, pretrained on 300B tokens from the Pile, significantly surpasses the vanilla Pythia-2.8B trained on the same data on both language modeling and a range of general downstream tasks. Furthermore, increasing the number of latent thoughts generated before each actual token—forming a chain analogous to CoT—consistently improves the model's performance. The code and models are available at https://github.com/LUMIA-Group/PonderLM-2.

## 1. Introduction

The conventional recipe for improving language models—scaling parameters and data—is yielding diminishing re-turns due to data scarcity (Villalobos et al., 2022; Muennighoff et al., 2023), scaling laws that predict diminishing returns (Hoffmann et al., 2022a; Hackenburg et al., 2025), and prohibitive communication overheads (Pati et al., 2023; Li et al., 2024).

This has shifted focus towards enhancing model capabilities via test-time scaling (Snell et al., 2024), particularly through methods based on Chain-of-Thought (CoT) (Jaech et al., 2024; DeepSeek-AI et al., 2025). CoT achieves remarkable success by generating long reasoning chains for each question, effectively scaling the generation steps and increasing computation per query. While effective, CoT relies on specialized datasets and complex training schemes (Allen-Zhu & Li, 2023; Li et al., 2025; Pang et al., 2025), is confined to a discrete token space, and is ultimately capped by the base model's capabilities (Yue et al., 2025).

An alternative direction is to scale computation during pre-training. One approach, often termed "*vertical scaling*", deepens the network by reusing parameters (Zeng et al., 2025; Giannou et al., 2023; Geiping et al., 2025; Chen et al., 2025). However, this can lead to training instabilities (Geiping et al., 2025) and often fails to outperform a standard dense model with a comparable inference budget, limiting its practical utility.

Inspired by the success of CoT in scaling generation steps, we propose a novel "*horizontal scaling*" approach: Pretraining Language Models with Latent Thoughts. Instead of deepening the model, our method teaches the LM to scale the generation process for each token. It first generates an intermediate latent thought—the last hidden state of the current position—which is then used as input to predict the actual subsequent token. This allows the model to refine its predictions in an unconstrained continuous space. To maintain training efficiency, we employ the Jacobi iteration (Saad, 2003; Barrett et al., 1994) to parallelize this inherently sequential process.

Our experiments show that, at an identical inference cost, a model trained with one latent thought per token surpasses a standard model with double the parameters. For instance, our PonderLM-2-1.4B models, built on Pythia and LLaMA architectures, significantly outperform their vanilla 2.8B counterparts trained on the same data. Our method also

---

[1]LUMIA Lab, School of Artificial Intelligence, Shanghai Jiao Tong University [2]Shanghai AI Laboratory [3]Shanghai Innovation Institute [4]Shanghai Jiao Tong University [5]Sun Yat-sen University. Correspondence to: Zhouhan Lin <lin.zhouhan@gmail.com>.

*Proceedings of the 43rd International Conference on Machine Learning*, Seoul, South Korea. PMLR 306, 2026. Copyright 2026 by the author(s).

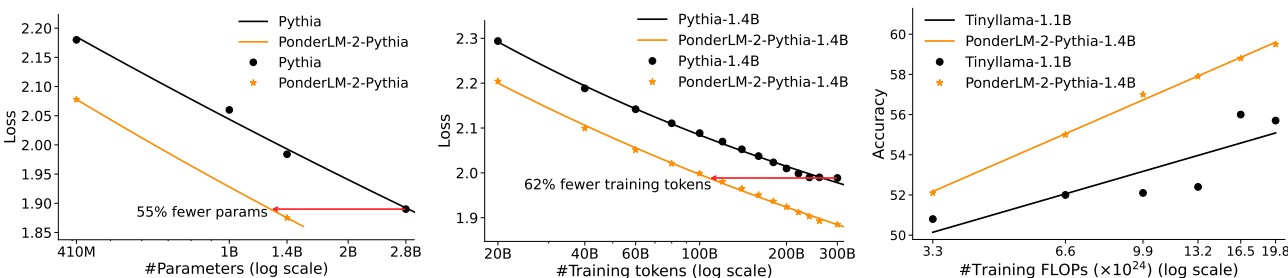

*Figure 1.* Scaling curves comparing PonderLM-2-Pythia with the official Pythia suite on the 300B Pile. Losses are reported on the held-out Pile validation split (test data). Our 1.26B model matches Pythia-2.8B with 55% fewer parameters (left), and our 1.4B model matches Pythia-1.4B with 62% fewer training tokens (middle). On 9 downstream tasks, PonderLM-2-Pythia attains higher average accuracy than TinyLlama at the same training FLOPs (right).

*Table 1.* A taxonomy of most related methods. The Computation Space column specifies where additional computation occurs. Our method is unique in its ability to learn a per-token, latent-space computational mechanism from a general corpus via a standard pretraining objective, without requiring specialized instruction data or complex training schemes like reinforcement learning.

| Method | Core Strategy | Training Data | Computation Space | Application Level | Training Method |
|---|---|---|---|---|---|
| CoT (Wei et al., 2022) | Scaling Generation | CoT Data | Explicit Token | Per Question | RL/SFT |
| Pause Tokens (Goyal et al., 2023) | Scaling Generation | General Corpus | Fixed Special Token | Per Token | Pretrain |
| Quiet-STaR (Zelikman et al., 2024) | Scaling Generation | General Corpus | Explicit Token | Per Token | RL |
| PonderLM (Zeng et al., 2025) | Scaling Model Depth | General Corpus | Continuous Embedding | Per Token | Pretrain |
| LoopedLM (Giannou et al., 2023) | Scaling Model Depth | General Corpus | Hidden State | Per Token | Pretrain |
| Coconut (Hao et al., 2024) | Scaling Generation | CoT Data | Hidden State | Per Question | SFT |
| PonderLM-2 (ours) | Scaling Generation | General Corpus | Hidden State | Per Token | Pretrain |

proves superior to previous vertical scaling techniques, even when their inference cost is twice as high. Furthermore, increasing the number of latent thoughts to form a chain of latent thoughts—analogous to CoT—before generating each real token consistently improves model performance, further underscoring the potential of our approach.

## 2. Related Work

We begin by discussing the two most related works: Coconut (Hao et al., 2024) and PonderLM (Zeng et al., 2025). Coconut *finetunes* a language model on *CoT data*, employing a "Chain of Continuous Thought"—represented by the final hidden states—to simulate explicit reasoning steps. This continuous chain is applied only after a question is posed. In contrast, our model learns this capability naturally during *pretraining* on a *general corpus*, appending a latent token after every token rather than just at the end of a prompt. PonderLM employs a *vertical* scaling strategy, deepening the model for a single generation step by iteratively re-feeding a "pondering embedding"—a probability-weighted sum of token embeddings—into its input layers. In contrast, our method utilizes a *horizontal* scaling approach. We extend the generative process for each token by appending latent thoughts, which are directly derived from the last hidden state of the previous computation step. A more comprehensive comparison with related works is provided in Table 1.

Other related methods (including test-time scaling and pa-

rameter sharing) can be broadly categorized into three main paradigms: scaling model depth via sequential parameter sharing, exploring multiple solutions through parallel computation, and scaling generation steps.

**Sequential Parameter Sharing to Scale Up Model Depth.** This paradigm increases a model's effective depth by reusing parameters. Early work such as Universal Transformers (Dehghani et al.) reused entire blocks; later methods iterate layers to refine hidden states (Geiping et al., 2025), feed outputs back as inputs (Giannou et al., 2023; Saunshi et al.), or recurrently apply a single layer to critical tokens (Chen et al., 2025). Despite gains, these approaches often incur inference overhead and training instability. In contrast, our horizontal scaling avoids deep recurrent computation by integrating thought into the sequence length.

**Exploring Multiple Solutions through Parallel Computation.** This paradigm generates multiple candidate solutions in parallel and selects the most promising one via a task-specific criterion. Representative examples include Best-of-$N$ sampling (Cobbe et al., 2021; Sun et al., 2024; Gui et al., 2024; Amini et al., 2024; Sessa et al., 2024) and Majority Voting (Wang et al., 2022). While effective for complex reasoning, these approaches are often computationally inefficient. Moreover, reliably identifying the best candidate remains challenging, since the verifier or selection heuristic may be suboptimal (Stroebl et al., 2024; Hassid et al., 2024).

**Scaling Generation Steps.** The most prominent method for

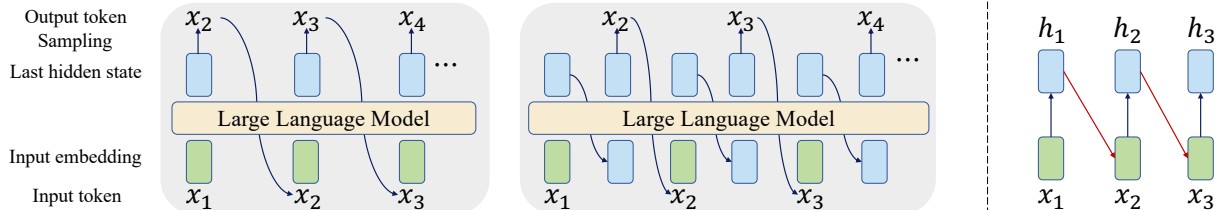

*Figure 2.* Comparison between standard LM and PonderLM-2. **Left:** standard autoregressive decoding, where each token is sampled after a single forward pass on the current prefix. **Middle:** PonderLM-2 inference, which delays token sampling and feeds the last hidden state back as the next-step input embedding, enabling latent thinking before emitting each token. **Right:** the resulting *sequential dependency* in a naive training unrolling. Red arrows indicate an *attention dependency*: computing $\mathbf{h}_i$ requires the prefix state $\mathbf{h}_{i-1}$ via causal attention, yielding $\mathbf{h}_1 \to \mathbf{h}_2 \to \cdots$ and preventing full parallelization. This motivates our Jacobi-iteration-based parallel training in Figure 3.

scaling generation steps is Chain-of-Thought (CoT) (Wei et al., 2022), which elicits reasoning paths from models before they provide a final answer. While effective, this process is often applied at the per-question level. Subsequent work has sought to integrate this "thinking" process more granularly into generation. One approach involves inserting non-content or "thinking" tokens into the sequence. For example, Goyal et al. (2023) inserted learnable "pause" tokens, while others explored discrete planning tokens (Wang et al., 2024) or filler tokens (Pfau et al., 2024). Quiet-STaR (Zelikman et al., 2024) even uses reinforcement learning to generate explicit rationale tokens between output tokens. However, these methods remain constrained to the discrete vocabulary space. In contrast, our work elevates this per-token computation into the continuous latent space.

## 3. Methodology

In this section, we introduce our pretraining method, which trains a language model to generate an intermediate latent thought before predicting the next token. For a standard Transformer LM, given an input sequence $x = (x_1, \ldots, x_T)$, we compute token embeddings $\mathbf{E}_t = [\mathbf{e}(x_1), \ldots, \mathbf{e}(x_t)]$ and obtain last-layer hidden states

$$\mathbf{H}_t = \text{Transformer}(\mathbf{E}_t), \quad (1)$$

where $\mathbf{e}(\cdot)$ is the embedding lookup. The hidden state at position $t$ is $\mathbf{h}_t = \mathbf{H}_t[t, :]$.

### 3.1. Inference Process

The inference process of our model is straightforward (Figure 2). For each token to be generated, the model first computes its corresponding last hidden state. This hidden state is then used as the next-step embedding, mimicking a recurrent thinking process.

### 3.2. Parallel Training via Jacobi Iteration

As illustrated in Figure 2, our inference introduces a recurrent feedback mechanism: after computing the last-layer hidden state at step $i$, we feed it back as the input embedding for generating step $i+1$. This induces an explicit

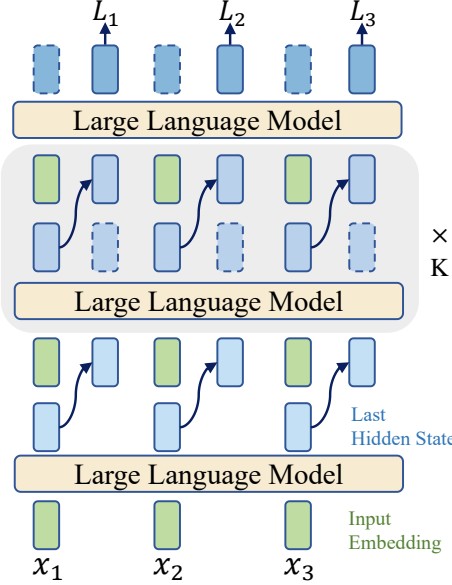

*Figure 3.* Parallel training procedure of our method via Jacobi iteration. **(1)** Compute initial hidden states from input embeddings $(x_1, x_2, x_3)$ and interleave them with the corresponding token embeddings to form a new sequence. **(2)** For $K$ rounds, all hidden states are updated in parallel. In each iteration, hidden states from the previous step are interleaved with the original embeddings to form the new input. To prevent depth overfitting, $K$ is *randomly sampled* from $\{2, 3\}$ per training instance (denoted by the dashed loop), rather than fixed. **(3)** Finally, the cross-entropy loss $(\mathcal{L}_1, \mathcal{L}_2, \mathcal{L}_3)$ is computed at the positions corresponding to the hidden state inputs to optimize language modeling.

left-to-right dependency (Figure 2, right): due to causal attention, computing $\mathbf{h}_i$ depends on the already-computed prefix states $\{\mathbf{h}_j\}_{j<i}$, so a naive token-wise unrolling would require $\mathbf{h}_1 \to \mathbf{h}_2 \to \cdots \to \mathbf{h}_T$, i.e., $T$ sequential forward passes for a length-$T$ sequence. Such a purely sequential training procedure is computationally infeasible for long contexts (e.g., $T{=}2048$). To enable efficient *parallel* training while remaining consistent with sequential inference, we approximate these autoregressive hidden states via **Jacobi iteration**, which refines all positions in parallel for a small number of rounds (Figure 3). The goal is to find a set of fixed-point hidden states $\mathbf{H}^* = [\mathbf{h}_1^*, \ldots, \mathbf{h}_T^*]$ that satisfy

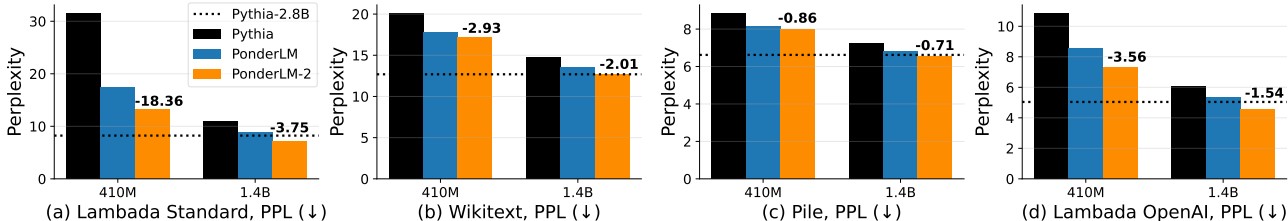

*Figure 4.* Language Modeling Perplexity (PPL). Our method achieves the lowest perplexity, consistently surpassing PonderLM despite its $2\times$ inference overhead at the same model size. Our PonderLM-2-Pythia-1.4B also outperforms the larger vanilla Pythia-2.8B. Numbers denote the absolute perplexity improvement ($\downarrow$) over the corresponding Pythia models.

a self-consistency condition: they are provided as part of the input and reproduced by the model as outputs. We solve for these states iteratively as follows:

**1. Initial Hidden State Estimation (Iteration** $0$**):** We first run a single forward pass on the original token embeddings $\mathbf{E} = [\mathbf{e}(x_1), \ldots, \mathbf{e}(x_T)]$ to obtain the initial hidden states:

$$[\mathbf{h}_1^0, \mathbf{h}_2^0, \ldots, \mathbf{h}_T^0] = \text{Transformer}([\mathbf{e}(x_1), \mathbf{e}(x_2), \ldots, \mathbf{e}(x_T)])$$

**2. Parallel State Update via Jacobi Iteration (Iteration** $k \to k+1$**):** For each subsequent iteration $k$, we construct a new input sequence by interleaving the original embeddings with the hidden states from the *previous* iteration, $\mathbf{H}^k$:

$$\mathbf{S}^k = [\mathbf{e}(x_1), \mathbf{h}_1^k, \mathbf{e}(x_2), \mathbf{h}_2^k, \ldots, \mathbf{e}(x_T), \mathbf{h}_T^k]$$

We then feed this sequence into the model to compute the updated hidden states for the next iteration, $\mathbf{H}^{k+1}$, in a single, parallel forward pass:

$$[\ldots, \mathbf{h}_1^{k+1}, \ldots, \mathbf{h}_2^{k+1}, \ldots] = \text{Transformer}(\mathbf{S}^k)$$

In this iterative process, all components of the new state vector $\mathbf{H}^{k+1}$ are computed in parallel based on the entire state vector from the previous iteration, $\mathbf{H}^k$. As shown in Figure 9, this iteration converges rapidly, with the hidden states stabilizing after a few rounds.

**3. Loss Computation:** After $K$ Jacobi iterations, we form the final input sequence $\mathbf{S}^K = [\mathbf{e}(x_1), \mathbf{h}_1^K, \ldots, \mathbf{e}(x_T), \mathbf{h}_T^K]$. The language modeling objective is then optimized by computing the cross-entropy loss $(L_1, L_2, \ldots, L_T)$ at the positions corresponding to the final hidden state inputs. Specifically, the loss $L_i$ is computed for predicting the token $x_{i+1}$ from the hidden state $\mathbf{h}_i^K$. To prevent overfitting to a fixed number of iterations, we randomly sample $K$ from $\{2, 3\}$ for each training instance, indicated by the dashed arrow in step (2) of Figure 3. As we show in Appendix E, this randomization is crucial: a fixed $K{=}2$ produces a pronounced spike at the first unseen step ($k{=}3$), indicating overfitting to a specific computational depth, whereas the randomized scheme yields a smooth, near-exponential decay that generalizes well beyond the trained range.

By formulating the training in this manner, we break the strict sequential dependency inherent in autoregressive models, thereby enabling efficient, parallel training.

### 3.3. Jacobi Iteration: Convergence and Consistency with Sequential Inference

Our parallel training can be formulated as a fixed-point iteration over hidden states. Given token embeddings $\mathbf{E} = [\mathbf{e}(x_1), \ldots, \mathbf{e}(x_T)] \in \mathbb{R}^{T \times d}$, let $\mathbf{H}^{(k)} = [\mathbf{h}_1^{(k)}, \ldots, \mathbf{h}_T^{(k)}] \in \mathbb{R}^{T \times d}$ denote the hidden states at Jacobi iteration $k$. In each iteration, we feed the interleaved sequence $\mathbf{S}^{(k)} = [\mathbf{e}(x_1), \mathbf{h}_1^{(k)}, \ldots, \mathbf{e}(x_T), \mathbf{h}_T^{(k)}]$ into the same masked Transformer and read out the updated states. Equivalently, this defines an operator $\Phi(\cdot; \mathbf{E})$ such that

$$\mathbf{H}^{(k+1)} = \Phi(\mathbf{H}^{(k)}; \mathbf{E}), \qquad \mathbf{H}^* = \Phi(\mathbf{H}^*; \mathbf{E}).$$

**Finite-step stabilization from causality.** By autoregressive causality, the computation of $\mathbf{h}_i$ depends only on the prefix $\{1, \ldots, i\}$. Hence, once $\{\mathbf{h}_1, \ldots, \mathbf{h}_{i-1}\}$ have stabilized, the update of $\mathbf{h}_i$ becomes fixed and will not change afterwards. Therefore, the iteration stabilizes left-to-right and reaches a fixed point in at most $T$ iterations.

**Consistency with sequential inference.** Let $\mathbf{H}_{\text{seq}}$ denote the hidden states produced by standard sequential autoregressive inference under the same model. At convergence, both $\mathbf{H}^*$ and $\mathbf{H}_{\text{seq}}$ satisfy the same causal update constraints, implying that the parallel objective is consistent with sequential inference. We empirically validate rapid convergence and alignment to $\mathbf{H}_{\text{seq}}$ in Section 4.6.

### 3.4. Position Embedding and Context Window

When feeding the last hidden state back as input, it preserves the original token's positional encoding. Thus the model's **usable context window does not shrink** (e.g., it is not "cut in half"): each token stays anchored to its original position.

# 4. Experiments

We conduct a comprehensive evaluation to validate both the effectiveness and efficiency of our method. We first scale up training on the 300B-token Pile to study scaling behavior and language modeling performance (Section 4.1). We then assess generalization on downstream benchmarks and instruction-following (Alpaca fine-tuning, MT-Bench) against strong open baselines (Section 4.2). To contextualize efficiency, we further compare with competitive compute-scaling methods (Looped Transformer, Pause Token, PonderLM) and a 2×-parameter oracle under matched and higher inference FLOPs budgets (Section 4.3). We also test plug-and-play effectiveness by continual pretraining an off-the-shelf LLaMA-3-3B foundation model (Section 4.4). Finally, we study complementarity with test-time scaling on GSM8K (Section 4.5) and provide convergence diagnostics of the Jacobi iteration (Section 4.6) together with ablations on key hyperparameters (Section 4.7).

## 4.1. Large-Scale Pretraining on Pile

We begin by validating our method at scale. We select the Pile (Gao et al., 2020), a substantial 300B-token dataset, as it provides a comprehensive pretraining corpus while remaining computationally tractable. We pretrain models based on the Pythia architecture (Biderman et al., 2023).

### 4.1.1. SCALING PROPERTIES

As illustrated in Figure 1, our pretrained models demonstrate superior scaling properties in both parameter and data efficiency, with all losses measured on the held-out Pile validation split (test data) rather than the training set. PonderLM-2-Pythia-1.26B, for instance, matches the performance of the official Pythia-2.8B with 55% fewer parameters. Furthermore, PonderLM-2-Pythia-1.4B converges to the official version's final performance using 62% less training data. Additional scaling curves on GPT-2 and LLaMA, presented in Appendix B, further validate the generalizability of our method.

### 4.1.2. LANGUAGE MODELING ABILITY

To further quantify these pretraining gains, we evaluate perplexity (PPL) on several standard benchmarks (Pile validation, Wikitext (Merity et al., 2016), and the Lambada (Paperno et al., 2016)). The results in Figure 4 show that our method delivers substantial and consistent PPL reductions across all model sizes and datasets. Notably, PonderLM-2-Pythia-1.4B is better than official Pythia-2.8B.

## 4.2. Downstream Task Evaluation

We now evaluate the practical capabilities of our previous pretrained Pythia models on a range of downstream tasks.

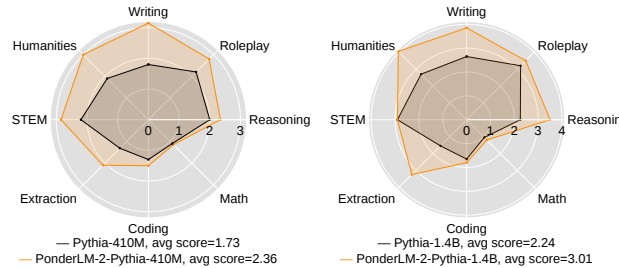

*Figure 5.* Instruction-following evaluation on MT-Bench. Our PonderLM-2-Pythia models outperform their official Pythia counterparts in all categories.

### 4.2.1. GENERAL DOWNSTREAM TASKS

**Datasets.** Following (Gu & Dao, 2023; Zeng et al., 2025), we including LAMBADA (Paperno et al., 2016), SciQ (Welbl et al., 2017), HellaSwag (Zellers et al., 2019), PIQA (Bisk et al., 2020), WinoGrande (Sakaguchi et al., 2021), ARC-Easy and ARC-Challenge (Clark et al., 2018), RACE (Lai et al., 2017) for comprehensive evaluation.

**Baselines.** We compare PonderLM-2-Pythia against: (1) the official Pythia suite; (2) prior PonderLM-Pythia models; (3) open-source models trained on comparable data volumes, including OPT (Zhang et al., 2022) and Bloom (Le Scao et al., 2023); and (4) TinyLLaMA (Zhang et al., 2024), trained on ~10× more data (3T vs. our 300B tokens).

**Results.** As shown in Table 2, our PonderLM-2-Pythia models consistently outperform similarly-sized baselines, including official Pythia, PonderLM-Pythia, OPT, and Bloom. Remarkably, our models also surpass competitors more than twice their size. For instance, PonderLM-2-Pythia-410M exceeds the performance of official Pythia-1B and Bloom-1.7B, while PonderLM-2-Pythia-1.4B outperforms Pythia-2.8B. Furthermore, our PonderLM-2-Pythia-1.4B significantly surpasses TinyLLaMA-1.1B, despite the latter being trained on 10× more data.

### 4.2.2. INSTRUCTION-FOLLOWING ABILITY

We assess instruction-following by fine-tuning the 410M and 1.4B versions of PonderLM-2-Pythia and the official Pythia models on Alpaca (Taori et al., 2023). Evaluated on MT-Bench (Zheng et al., 2023), our pretrained PonderLM-2-Pythia consistently outperforms the corresponding Pythia baselines across all categories (see Figure 5), improving the average score by 0.63 (410M) and 0.77 (1.4B).

## 4.3. Comparison with Baseline Methods

To contextualize the performance and efficiency of our proposed method, we conduct a detailed comparison against several competitive baselines on the LLaMA architecture.

*Table 2.* Zero-shot and five-shot accuracy (%) on downstream tasks. All pretrained model weights used for comparison are obtained from their official repositories. Δacc indicates the average accuracy improvement over the corresponding Pythia baseline. Italicized models are shown but not bolded, since they use significantly larger training data or parameters, and their avg acc are marked in red when outperformed by our model. Ponder refers to the PonderLM-Pythia model, whose inference cost is *twice* ours under the same parameter size, with results taken from their original paper.

| Model (#training tokens) | Lambada OpenAI | ARC -E | Lambada Standard | ARC -C | Wino Grande | PIQA | Hella Swag | SciQ | RACE | Avg acc / Δacc ↑ |
|---|---|---|---|---|---|---|---|---|---|---|
| **0-shot** | | | | | | | | | | |
| Pythia-410M (300B) | 51.4 | 52.2 | 36.4 | 21.4 | 53.8 | 66.9 | 33.7 | 81.5 | 30.9 | 47.6 |
| OPT-350M (300B) | 45.2 | 44.0 | 35.8 | 20.7 | 52.3 | 64.5 | 32.0 | 74.9 | 29.8 | 44.4 |
| Bloom-560M (366B) | 34.3 | 47.5 | 33.3 | 22.4 | 51.5 | 63.8 | 31.5 | 80.3 | 30.5 | 43.9 |
| Ponder-410M (300B) | 56.9 | 51.9 | 45.3 | 22.6 | **56.0** | 68.7 | 37.0 | 81.4 | **33.8** | 50.4 |
| *Pythia-1B (300B)* | *55.9* | *56.8* | *42.0* | *24.2* | *52.5* | *70.5* | *37.7* | *83.3* | *32.7* | *50.6* |
| **PonderLM-2-Pythia-410M** ( 300B) | **59.1** | **54.0** | **47.3** | **24.6** | 55.5 | 69.4 | 37.7 | **86.2** | 33.5 | **51.9** / +4.3 |
| Pythia-1.4B (300B) | 61.6 | 60.4 | 49.7 | 25.9 | 57.5 | 70.8 | 40.4 | 86.4 | 34.1 | 54.1 |
| OPT-1.3B (300B) | 57.9 | 57.1 | 52.5 | 23.4 | 59.7 | 71.8 | 41.6 | 84.3 | 34.3 | 53.6 |
| Bloom-1.7B (366B) | 46.2 | 56.4 | 44.5 | 23.7 | 56.8 | 68.5 | 37.5 | 85.0 | 33.2 | 50.2 |
| Ponder-1.4B (300B) | 65.2 | 62.0 | 53.8 | 27.0 | 60.1 | 72.6 | 44.0 | 89.0 | 35.2 | 56.5 |
| *Tinyllama-1.1B (3T)* | *58.8* | *60.3* | *49.3* | *28.0* | *59.0* | *73.3* | *45.0* | *88.9* | *36.4* | *55.4* |
| *Pythia-2.8B (300B)* | *64.6* | *64.4* | *54.3* | *29.5* | *60.2* | *73.8* | *45.4* | *88.5* | *34.9* | *57.3* |
| **PonderLM-2-Pythia-1.4B** (300B) | **67.6** | **64.1** | **57.1** | **30.2** | **60.9** | 72.9 | **45.8** | **91.0** | **37.1** | **58.5** / +4.4 |
| **5-shot** | | | | | | | | | | |
| Pythia-410M (300B) | 43.9 | 54.7 | 32.8 | 22.3 | 53.4 | 68.0 | 33.8 | 88.9 | 30.4 | 47.6 |
| OPT-350M (300B) | 38.3 | 45.4 | 32.1 | 20.5 | 53.0 | 65.8 | 31.9 | 85.7 | 29.5 | 44.7 |
| Bloom-560M (366B) | 29.4 | 50.2 | 29.7 | 21.9 | 52.7 | 64.2 | 31.4 | 88.0 | 30.0 | 44.2 |
| Ponder-410M (300B) | 48.9 | **58.7** | 43.7 | **26.1** | 54.0 | 70.5 | 37.3 | 91.0 | 32.4 | 51.4 |
| *Pythia-1B (300B)* | *48.3* | *58.6* | *35.8* | *25.4* | *52.8* | *71.3* | *37.7* | *91.6* | *31.7* | *50.4* |
| **PonderLM-2-Pythia-410M** (300B) | **52.1** | 58.0 | **45.0** | 26.0 | **54.6** | 69.2 | **37.9** | **91.7** | **32.6** | **51.9** / +4.3 |
| Pythia-1.4B (300B) | 54.5 | 63.1 | 44.5 | 28.8 | 57.1 | 71.0 | 40.5 | 92.4 | 34.6 | 54.1 |
| OPT-1.3B (300B) | 54.0 | 60.4 | 49.0 | 26.9 | 56.9 | 72.4 | 38.5 | 91.8 | 35.4 | 52.7 |
| Bloom-1.7B (366B) | 42.5 | 58.8 | 41.5 | 26.2 | 57.7 | 68.7 | 37.6 | 91.9 | 33.5 | 50.9 |
| Ponder-1.4B (300B) | 59.2 | **67.5** | 49.9 | 32.4 | 60.4 | 73.5 | 44.2 | 94.3 | 37.1 | 57.6 |
| *Tinyllama-1.1B (3T)* | *53.8* | *64.8* | *45.0* | *31.1* | *59.4* | *73.8* | *44.9* | *94.0* | *36.4* | *55.9* |
| *Pythia-2.8B (300B)* | *59.0* | *67.0* | *50.7* | *31.0* | *61.1* | *74.4* | *45.3* | *93.7* | *35.9* | *57.6* |
| **PonderLM-2-Pythia-1.4B** (300B) | **63.6** | 67.4 | **56.0** | **32.6** | **64.0** | 73.5 | **46.4** | **94.5** | **37.9** | **59.5** / +5.4 |

**Baselines.** We compare against three compute-scaling baselines and a parameter-matched oracle: **Looped Transformer** (Saunshi et al.), which repeats the full transformer stack for multiple loops; **Pause Token** (Goyal et al., 2023), which prepends a fixed number of learnable pause tokens before each generated token; **Pondering LLM** (Zeng et al., 2025), which iteratively feeds back a probability-weighted "pondering embedding" for refinement; and a **Scaled-up Model** oracle, a standard LLaMA with *doubled depth* (doubling the number of layers to reach ∼2.8B parameters), whose inference FLOPs are comparable to our method. For iterative baselines (Looped Transformer, Pause Token, and PonderLM), we consider two compute budgets. First, we match our inference FLOPs (a 2× increase over the vanilla model), using 2 loops, 1 pause token, or 1 pondering step. Second, for a stronger baseline, we pretrain them with substantially higher inference FLOPs than our method.

**Settings.** We use LLaMA-1.4B as the testbed. All models are trained on 26B tokens with identical hyperparameters

for fair comparison. We report perplexity on Pile validation, WikiText, and LAMBADA (OpenAI and standard), along with average accuracy over the nine downstream tasks. Computational overhead is measured as relative inference FLOPs w.r.t. the vanilla 1.4B model.

**Training Computation Analysis.** We compare training FLOPs under the same data budget. Baselines with a 4× inference budget (e.g., 4 loops / 3 pause tokens / 3 pondering steps) incur roughly 4× training FLOPs, while the 2.8B scaled model costs about 2× due to doubled parameters. Our training consists of: (i) one forward pass on the original sequence (1×), (ii) $K$ Jacobi iterations on an interleaved token–thought sequence (doubling length, 2× per iteration), and (iii) a final forward pass on the interleaved sequence (2×). Thus the training FLOPs multiplier is $1 + 2K + 2 = 3 + 2K$. With $K \in \{2, 3\}$ and $\mathbb{E}[K] = 2.5$, the average cost is $\approx 3 + 2 \times 2.5 = 8\times$ the vanilla baseline.

**Results.** *Training-compute lens.* Although our method has

*Table 3.* Comparison on various benchmarks. Inference FLOPs are relative to the vanilla model. Actual throughput is evaluated following (Wu & Tu, 2024). Our method shows superior performance across all metrics while maintaining inference efficiency. Detailed downstream tasks performance are provided in Appendix A.

| Model | Inference FLOPs | Throughput (tokens/s) | Pile | Lambada OpenAI | Wikitext | Lambada Standard | Avg Acc 0 shot | Avg Acc 5 shot |
|---|---|---|---|---|---|---|---|---|
| LLaMA-1.4B (train from scratch) | $1\times$ | 221.19 | 9.04 | 11.42 | 20.18 | 27.52 | 47.7 | 47.5 |
| **Methods with comparable ($2\times$) inference FLOPs** | | | | | | | | |
| Looped LLaMA-1.4B (2 loops) | $2\times$ | 112.33 | 8.35 | 9.22 | 18.34 | 21.50 | 49.8 | 48.5 |
| Pause LLaMA-1.4B (1 pause) | $2\times$ | 112.57 | 8.58 | 9.87 | 19.10 | 19.90 | 48.5 | 48.2 |
| Pondering LLaMA-1.4B (1 step) | $2\times$ | 110.16 | 8.33 | 9.26 | 18.36 | 19.95 | 49.6 | 49.2 |
| LLaMA-2.8B (train from scratch) | $2\times$ | 110.50 | 8.23 | 8.93 | 18.09 | 17.08 | 49.8 | 50.6 |
| **PonderLM-2-LLaMA-1.4B** | $2\times$ | 111.55 | **7.89** | **7.39** | **16.99** | **12.20** | **52.3** | **51.9** |
| **Methods with higher inference FLOPs** | | | | | | | | |
| Looped LLaMA-1.4B (4 loops) | $4\times$ | 59.48 | 8.04 | 8.14 | 17.35 | 15.14 | 50.9 | 50.5 |
| Pause LLaMA-1.4B (3 pauses) | $4\times$ | 60.43 | 8.17 | 7.92 | 17.81 | 13.76 | 51.0 | 50.4 |
| Pondering LLaMA-1.4B (3 steps) | $4\times$ | 55.42 | 8.03 | 8.02 | 17.23 | 15.48 | 51.5 | 51.5 |
| Pondering LLaMA-1.4B (7 steps) | $8\times$ | 26.39 | 8.01 | 7.94 | 17.21 | 14.23 | 51.4 | 51.5 |

*Table 4.* 0-shot and 5-shot accuracy (%) on downstream tasks, evaluating LLaMA-3-3B enhancement via continual pre-training (CPT). The table compares three models: original LLaMA-3-3B, vanilla CPT, and our method with continual pre-training.

| Model | Lambada OpenAI | ARC -E | Lambada Standard | ARC -C | Wino Grande | PIQA | Hella Swag | SciQ | RACE | Avg acc / $\Delta$acc $\uparrow$ |
|---|---|---|---|---|---|---|---|---|---|---|
| **0-shot** | | | | | | | | | | |
| LLaMA-3-3B | 70.1 | 74.5 | 63.7 | 42.2 | 69.0 | 76.8 | 55.4 | 95.5 | 39.4 | 65.2 |
| Standard CPT | 69.4-0.7 | 76.2+1.7 | 65.1+1.4 | 42.5+0.3 | 67.5-1.5 | 77.2+0.4 | 55.1-0.3 | 94.1-1.4 | 39.6+0.2 | 65.2 |
| **PonderLM-2 CPT** | 70.8+0.7 | 76.3+1.8 | 67.5+3.8 | 42.1-0.1 | 69.0+0.0 | 77.9+1.1 | 56.1+0.7 | 94.6-0.9 | 40.5+1.1 | **66.2** |
| **5-shot** | | | | | | | | | | |
| LLaMA-3-3B | 66.8 | 78.1 | 64.1 | 44.1 | 71.4 | 78.6 | 56.1 | 96.4 | 41.8 | 66.4 |
| Standard CPT | 66.1-0.7 | 77.7-0.4 | 66.0+1.9 | 43.0-1.1 | 71.5+0.1 | 78.3-0.3 | 55.8-0.3 | 96.5+0.1 | 40.6-1.2 | 66.2 |
| **PonderLM-2 CPT** | 69.0+2.2 | 78.6+0.5 | 67.4+3.3 | 48.0+3.9 | 72.9+1.5 | 78.1-0.5 | 57.0+0.9 | 96.7+0.3 | 41.4-0.4 | **67.7** |

higher training FLOPs when trained from scratch, it delivers better performance per training compute. Under comparable training FLOPs, we outperform PonderLM even with 7 pondering steps, and our 1.4B model achieves consistently higher mean downstream accuracy than TinyLLaMA-1.1B at matched training FLOPs despite TinyLLaMA using 3T tokens (Figure 1, right). Furthermore, this overhead can be largely mitigated by continual pretraining from existing checkpoints, where we observe consistent gains on off-the-shelf foundation models (Section 4.4). *Inference-compute lens.* The results in Table 3 further show that our method achieves the lowest perplexity across all language modeling benchmarks and the highest average downstream accuracy under matched inference FLOPs. Notably, at the comparable $2\times$ inference budget, we surpass all baselines including the $2\times$-parameter (doubled-depth) LLaMA-2.8B oracle, and we also maintain a clear advantage over methods operating at higher ($4\times$ and beyond) inference budgets.

### 4.4. Effectiveness on off-the-shelf Foundation Models

We further examine whether our method can improve existing large foundation models. Specifically, we start from the official LLaMA-3-3B (Grattafiori et al., 2024) and conduct continual pre-training on 5B tokens from SlimPajama (Shen et al., 2024). We compare the original model with a vanilla continual pre-training baseline and our approach. As shown in Figure 6, our method attains lower training loss after consuming fewer than 1B tokens, and the gap continues to grow with training. On our nine standard downstream tasks (Table 4), it yields a clear performance gain, highlighting its plug-and-play value for off-the-shelf models.

### 4.5. Complementing with Test-Time Scaling

We verify whether our method *complements* test-time scaling (TTS). We reuse the two continual-pretrained models from Section 4.4: Standard CPT and PonderLM-2 CPT, both initialized from LLaMA-3-3B and trained on the same SlimPajama stream. We evaluate on GSM8K using Exact Match (EM). For TTS, we apply Majority Voting and Best-of-$N$ with $N \in \{1, \ldots, 10\}$ samples, and additionally test Chain-of-Thought (CoT) prompting. Figure 8 shows that both Voting and Best-of-$N$ improve as $N$ increases, while PonderLM-2 CPT consistently achieves higher EM and typically obtains larger gains than Standard CPT. Table 5 further

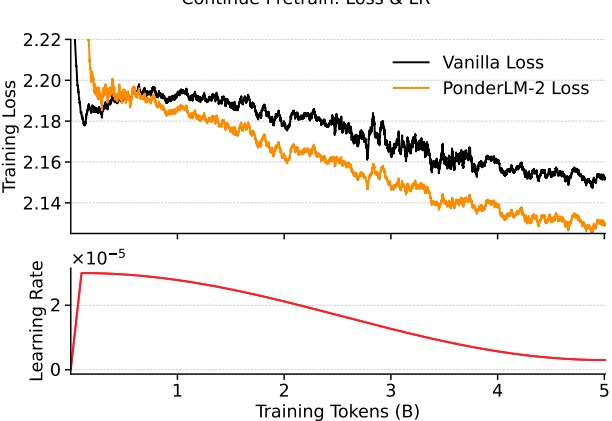

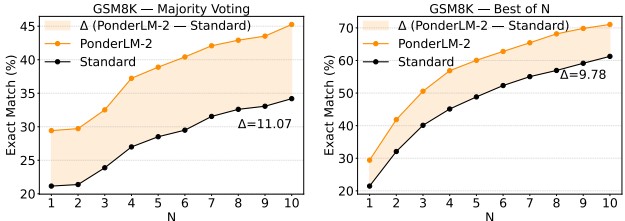

*Figure 6.* Training loss of PonderLM-2 CPT and vanilla continual pretraining (CPT), both starting from the official LLaMA-3-3B and trained on 5B tokens from SlimPajama with peak LR = 3e−5. The PonderLM-2 curve exhibits a small initial spike as the model accommodates the newly introduced latent states, then drops below the vanilla curve within ∼1B tokens. A peak-LR ablation flanking this setting (1e−5 and 5e−5) confirms the same pattern; see Appendix M.

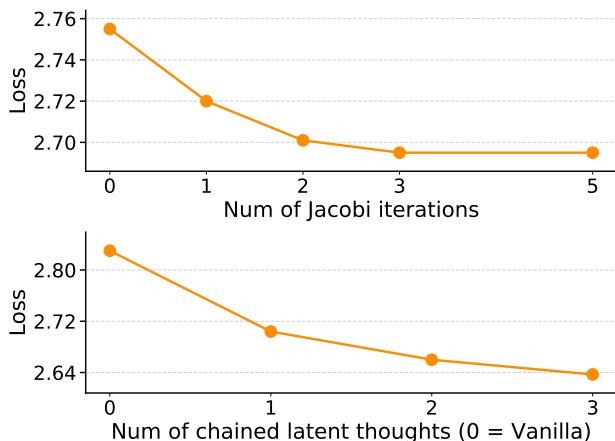

*Figure 7.* Ablations on Pythia-70M trained on 30B tokens from the Pile. (Top) Impact of the number of Jacobi iterations $K$. (Bottom) Chaining more latent thoughts before each token lowers held-out validation loss. A larger-scale check on LLaMA-410M (6B tokens) with $K$ up to 5 is reported in Appendix G.

*Figure 8.* GSM8K Exact Match (EM) under two test-time scaling strategies on the LLaMA-3-3B continual-pretraining models from Section 4.4 (5B SlimPajama tokens): Majority Voting (left) and Best-of-$N$ (right), with $N \in \{1, \dots, 10\}$ sampled solutions per problem. EM increases with $N$ for both methods, while PonderLM-2 CPT consistently outperforms Standard CPT and typically yields larger gains. A compute-matched view of the same comparison, where PonderLM-2 with $N$ samples is matched against Standard CPT with $2N$ samples (since one PonderLM-2 sample costs ∼2× the inference FLOPs of a Standard CPT sample), is provided in Appendix O.

*Table 5.* CoT gains on GSM8K. Exact Match (%) with and without CoT for Standard CPT and PonderLM-2 CPT. CoT improves both models, and the improvement is larger for PonderLM-2 CPT.

| Setting | Standard CPT | PonderLM-2 CPT |
|---|---|---|
| Without CoT | 10.01 | **12.59** |
| With CoT | 24.26+14.25 | **32.90**+20.31 |

shows that CoT boosts both models, with PonderLM-2 CPT benefiting more, indicating that our method is complementary to explicit reasoning prompting.

### 4.6. Convergence Diagnostics of Jacobi Iteration

We evaluate the convergence of Jacobi updates in parallel training by tracking (i) the per-iteration change $r_k$ between successive iterates and (ii) the distance $d_k$ to the sequential autoregressive hidden states $H_{\text{seq}}$ (both measured by RMSE; see Figure 9). **Convergence to a fixed point:** As shown in Figure 9 (Left), $r_k$ drops rapidly and is approximately

linear on a semi-log scale in early iterations, indicating near-exponential decay, before saturating at the BFloat16 numerical floor. **Agreement with sequential inference:** Figure 9 (Right) shows that $d_k$ reaches the same floor within a few iterations, suggesting that the Jacobi iterates match the sequential solution up to numerical precision.

### 4.7. Ablation Study

In this section, we study the impact of key components. We ablate the number of Jacobi iterations, different position embedding strategies, and the number of latent thoughts (chain analogous to CoT) generated before each token. All experiments are conducted on Pythia-70m with 30B tokens.

As shown in Figure 7 (Top), increasing the number of Jacobi iterations initially lowers the loss, but the improvement saturates after only 3 iterations, which corroborates the fast convergence we observe (Figure 9). Meanwhile, as illustrated in Figure 7 (Bottom), chaining more latent thoughts consistently leads to better performance, which demonstrates our method's potential for pretraining LLMs to generate a chain of latent thoughts before predicting each token. Regarding

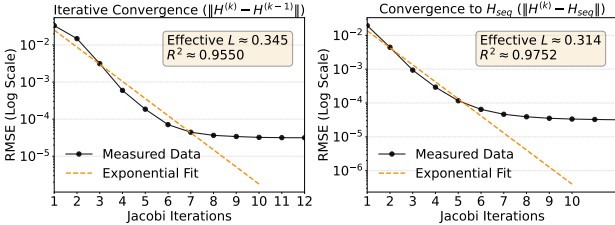

*Figure 9.* Empirical convergence of Jacobi iteration on our PonderLM-2-Pythia-410M pretrained on 300B tokens from the Pile. (Left) RMSE between adjacent iterates $r_k = \mathrm{RMSE}(H^{(k)} - H^{(k-1)})$ rapidly decays and then saturates at the BFloat16 floor. (Right) RMSE to sequential inference $d_k = \mathrm{RMSE}(H^{(k)} - H_{\mathrm{seq}})$ quickly drops to the same floor, showing agreement with the sequential result. We report an empirical decay rate $L$ (the slope of a log-linear fit over early, pre-saturation iterations) and the corresponding goodness-of-fit $R^2$. A complementary measurement on the larger LLaMA-3-3B CPT model is provided in Appendix N.

position embedding, we compared assigning sequential position ids to thoughts versus reusing the token's position id for its corresponding thoughts and found a negligible performance difference. We use the latter strategy to avoid the need for long-context capabilities.

## 5. Conclusion

In this paper, we introduce the approach of pretraining language models with latent thoughts, which can be effectively realized using large-scale general corpora. Language models pretrained with latent thoughts consistently outperforms its counterparts with double the parameters (at equal inference cost), as well as prior related methods like PonderLM, looped, and paused models, even when they use double the inference budget. Furthermore, we show that chaining latent thoughts, akin to COT, consistently improves model performance. We posit that our work introduces a new potential dimension for scaling the capabilities of language models.

## Impact Statement

This paper presents work whose goal is to advance the field of Machine Learning. There are many potential societal consequences of our work, none which we feel must be specifically highlighted here.

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

## A. Detailed Downstream Tasks Performance

*Table 6.* Zero-shot and five-shot accuracy (%) on downstream tasks, as described in Section 4.3.

| Model | Lambada OpenAI | ARC -E | Lambada Standard | ARC -C | Wino Grande | PIQA | Hella Swag | SciQ | RACE | Avg acc / $\Delta$acc ↑ |
|---|---|---|---|---|---|---|---|---|---|---|
| **0-shot** | | | | | | | | | | |
| LLaMA-1.4B (train from scratch) | 50.6 | 52.2 | 37.1 | 20.9 | 53.0 | 65.6 | 33.6 | 84.5 | 31.8 | 47.7 |
| **Methods with comparable ($2\times$) inference FLOPs** | | | | | | | | | | |
| Looped LLaMA-1.4B (2 loops) | 53.6 | 54.3 | 41.8 | 23.6 | 52.9 | 69.3 | 35.8 | 83.6 | **33.5** | 49.8 |
| Pause LLaMA-1.4B (1 pause) | 51.8 | 53.8 | 40.1 | 21.8 | 52.5 | 67.5 | 34.7 | 83.1 | 30.9 | 48.5 |
| Pondering LLaMA-1.4B (1 step) | 53.8 | 53.2 | 42.6 | 23.0 | 52.6 | 68.4 | 35.9 | 83.2 | 33.3 | 49.6 |
| LLaMA-2.8B (train from scratch) | 54.3 | 53.4 | 43.5 | 24.4 | 53.1 | 68.0 | 36.2 | 83.4 | 31.5 | 49.8 |
| **PonderLM-2-LLaMA-1.4B** | **58.1** | **58.0** | **48.2** | **25.2** | **53.9** | **70.7** | **38.6** | **85.9** | 32.4 | **52.3**/+4.6 |
| **Methods with higher ($4\times$) inference FLOPs** | | | | | | | | | | |
| Looped LLaMA-1.4B (4 loops) | 55.8 | 55.2 | 45.6 | 23.2 | 54.1 | 68.9 | 37.6 | 84.9 | 33.0 | 50.9 |
| Pause LLaMA-1.4B (3 pauses) | 56.2 | 54.0 | 46.5 | 24.2 | 55.3 | 68.8 | 36.7 | 85.4 | 32.3 | 51.0 |
| Pondering LLaMA-1.4B (3 step) | 56.7 | 56.3 | 45.4 | 23.8 | 55.6 | 68.3 | 37.8 | 86.3 | 33.0 | 51.5 |
| **5-shot** | | | | | | | | | | |
| LLaMA-1.4B (train from scratch) | 45.1 | 53.5 | 34.7 | 22.3 | 50.9 | 66.1 | 33.6 | 89.3 | **31.6** | 47.5 |
| **Methods with comparable ($2\times$) inference FLOPs** | | | | | | | | | | |
| Looped LLaMA-1.4B (2 loops) | 46.0 | 55.6 | 36.9 | 23.9 | 51.1 | 69.2 | 35.7 | 90.1 | 27.9 | 48.5 |
| Pause LLaMA-1.4B (1 pause) | 45.5 | 56.1 | 35.6 | 23.8 | 50.8 | 68.0 | 34.9 | 88.4 | 30.7 | 48.2 |
| Pondering LLaMA-1.4B (1 step) | 48.0 | 56.3 | 40.9 | 24.4 | 53.3 | 69.2 | 36.2 | 89.5 | 25.0 | 49.2 |
| LLaMA-2.8B (train from scratch) | 49.7 | 57.2 | 43.7 | 25.3 | 53.1 | 69.3 | 36.3 | 89.6 | 30.8 | 50.6 |
| **PonderLM-2-LLaMA-1.4B** | **49.8** | **59.6** | **45.6** | **27.7** | **56.3** | **69.8** | **38.7** | **91.3** | 28.0 | **51.9** /+4.4 |
| **Methods with higher ($4\times$) inference FLOPs** | | | | | | | | | | |
| Looped LLaMA-1.4B (4 loops) | 48.0 | 58.5 | 42.8 | 25.0 | 54.8 | 70.4 | 37.6 | 89.2 | 28.5 | 50.5 |
| Pause LLaMA-1.4B (3 pauses) | 48.8 | 56.6 | 41.0 | 24.1 | 54.1 | 69.3 | 36.6 | 90.7 | 32.6 | 50.4 |
| Pondering LLaMA-1.4B (3 step) | 49.5 | 58.8 | 42.6 | 25.4 | 54.3 | 69.2 | 37.9 | 91.2 | 34.7 | 51.5 |

## B. Generalization to GPT-2 and LLaMA

To verify the general applicability of our method, we apply our proposed latent mechanism to the widely-used GPT-2 and LLaMA architectures.

**Experimental Settings.** We train both vanilla and latent-enhanced versions of these models from scratch on a subset of the Pile dataset, with sizes ranging from 405M to 1.4B parameters. The experimental setup, including the number of training tokens aligned with Chinchilla scaling laws (Hoffmann et al., 2022b). Detailed model configurations and training hyperparameters are provided in Table 7.

**Results.** The scaling curves are presented in Figure 10. Our method provides significant and consistent performance improvements for both GPT-2 and LLaMA across all model sizes. Notably, PonderLM-2-GPT-834M and PonderLM-2-LLaMA-834M achieve a validation loss comparable to their vanilla counterparts trained with approximately **2.48x** and **2.46x** the parameter-token product, respectively.

*Table 7.* Model sizes and hyperparameters for the scaling experiments on GPT-2 and LLaMA.

| Parameters | $n_{layers}$ | $d_{model}$ | $n_{heads}$ | Learning Rate | Batch Size (tokens) | Training Tokens |
|---|---|---|---|---|---|---|
| 405M | 24 | 1024 | 16 | 3.0e-4 | 0.5M | 7B |
| 834M | 24 | 1536 | 24 | 2.5e-4 | 0.5M | 15B |
| 1.4B | 24 | 2048 | 32 | 2.0e-4 | 0.5M | 26B |

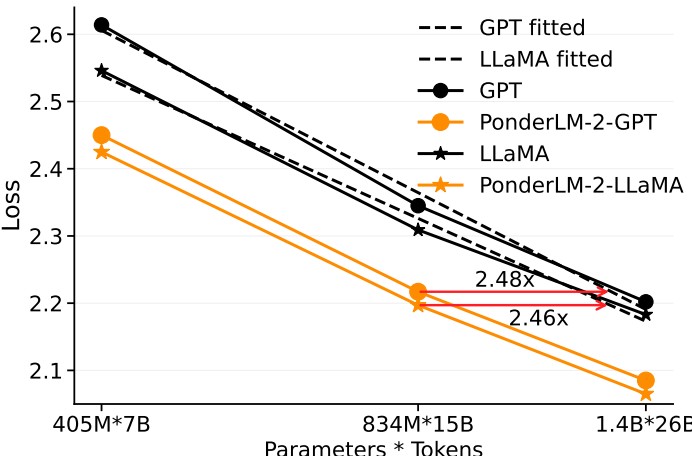

*Figure 10.* Scaling curve comparison on the Pile (held-out validation loss): vanilla GPT-2 and LLaMA backbones against our PonderLM-2 variants. Model sizes range from 405M to 1.4B parameters; the number of training tokens at each size follows the Chinchilla-style schedule of Table 7 (7B for 405M, 15B for 834M, 26B for 1.4B). Equivalently, each point corresponds to a fixed training-FLOPs budget; PonderLM-2 matches a vanilla model trained with ∼2.5× the parameter–token product at the same FLOPs.

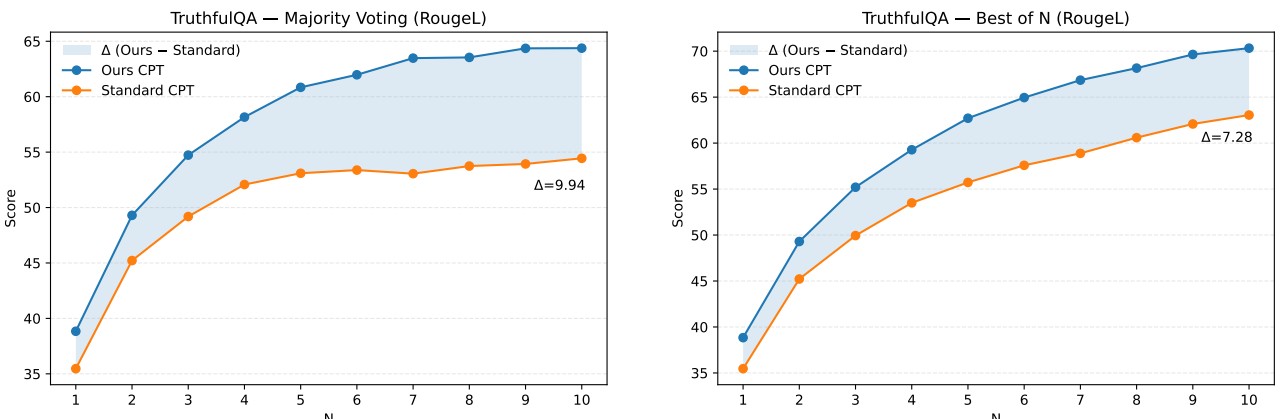

*Figure 11.* Test-time scaling on TRUTHFULQA. Majority Voting (left) and Best-of-$N$ (right). PonderLM-2 CPT benefits more than Standard CPT across $N$.

## C. Complementing Models with Test-Time Scaling Approaches

We study whether our method (denoted **PonderLM-2 CPT**) complement common test-time scaling strategies, compared with a vanilla baseline (**Standard CPT**). We evaluate on TRUTHFULQA (ROUGE-L) and GSM8K (Exact Match), and set the number of samples $N \in \{1, \ldots, 10\}$ for Best-of-$N$ and Majority Voting.

**Settings.** We reuse the two models from Section 4.4: both start from the official LLaMA-3-3B backbone and undergo continual pretraining on the same SlimPajama dataset. One is a vanilla continual-pretraining baseline (Standard CPT), while the other incorporates our latent thoughts during continual pretraining (PonderLM-2 CPT). We evaluate three test-time strategies: Majority Voting, Best-of-$N$ (BoN), and Chain-of-Thought (CoT) prompting, varying only the number of samples $N$ for voting and BoN.

### C.1. Majority Voting

As $N$ increases, both models improve, but the gap between them widens consistently on TRUTHFULQA and GSM8K. This pattern suggests that PonderLM-2 CPT produces batches of answers that are clustered and consistently correct, making aggregation particularly effective. (See Figure 8 (left) and Figure 11 (left).)

## C.2. Best of N

BoN likewise amplifies PonderLM-2 CPT over Standard CPT and improves monotonically with $N$, indicating that our model reliably produces a *diverse set of high-quality candidates* from which a strong single answer can be selected. While its gains are typically a bit smaller than Majority Voting at the same $N$, the strong BoN curve is nevertheless evidence against mode collapse: PonderLM-2 CPT generates multiple plausible solutions, and either selecting the best (BoN) or aggregating them (voting) yields consistent benefits. (See Figure 8 (right) and Figure 11 (right).)

## C.3. CoT

We further test CoT prompting on GSM8K. As summarized in Table 5, CoT improves both models, with PonderLM-2 CPT benefiting more. This suggests a complementary relationship between the two techniques, where their combination leads to more reliable outcomes.

# D. Jacobi Convergence Analysis and Equivalence to Sequential Inference

In this section, we provide a rigorous theoretical and empirical analysis to demonstrate that the parallel Jacobi iteration employed during training is mathematically consistent with standard sequential inference. We structure our analysis along a four-step logical chain:

1. **Convergence:** The parallel iteration is theoretically guaranteed to converge to a fixed point $H^*$;
2. **Exponential Rate:** The rapid decay of iterative updates proves that the convergence is exponential;
3. **Trajectory Alignment:** Empirical evidence shows the iteration also converges to the sequential solution $H_{\text{seq}}$;
4. **Equivalence:** By the uniqueness of limits, it follows that $H^* = H_{\text{seq}}$, proving the solutions are identical.

## D.1. Exact Convergence Guarantee of Parallel Training (Finite-Step Property)

We view the Jacobi iteration in parallel training as a process of finding a fixed point. Let $E \in \mathbb{R}^{T \times d}$ be the fixed input Embeddings, and $H^{(k)} \in \mathbb{R}^{T \times d}$ be the hidden states at iteration $k$. We treat the Transformer layers as a non-linear operator $\Phi$, with the update rule $H^{(k+1)} = \Phi(H^{(k)}; E)$. Our goal is to find the fixed point $H^*$ satisfying $H^* = \Phi(H^*; E)$.

Unlike general fixed-point problems, the Autoregressive Causality of the Transformer guarantees stability. Specifically, since the computation of the $i$-th token depends strictly on preceding tokens ($j < i$), convergence follows a clear inductive chain: the first token stabilizes immediately based on the fixed input, and subsequently, any token $k$ stabilizes once its preceding context (tokens 1 to $k-1$) is fixed.

This guarantees that for a sequence of length $T$, the entire sequence strictly converges to the fixed point $H^*$ in at most $T$ steps.

## D.2. Fast Exponential Convergence

While the above theorem provides a "worst-case" guarantee (taking $T$ steps), the core advantage of our method is that its convergence speed is significantly faster than $T$. We introduce the Banach Fixed-Point Theorem for analysis.

**Theoretical Analysis:**

- **Definition:** According to the Banach Fixed-Point Theorem, if there exists a Lipschitz constant $L$ ($0 \leq L < 1$) under some norm $\| \cdot \|$, such that for any $H_a, H_b$, $\|\Phi(H_a) - \Phi(H_b)\| \leq L\|H_a - H_b\|$ holds, then the iteration must converge to a unique fixed point $H^*$.
- **Convergence Speed:** If $L < 1$, the error at iteration $k$ will decay exponentially: $\|H^{(k)} - H^*\| \leq L^k\|H^{(0)} - H^*\|$. This indicates that the algorithm possesses an exponential convergence property, meaning the error magnitude shrinks by a fixed ratio $L$ at each step.
- **Posterior Error Estimation (Cauchy Property):** Since the true fixed point $H^*$ is unknown, we rely on the property of Cauchy Sequences. The upper bound of the error to the true solution is determined by the RMSE of adjacent iterations:

$$\|H^{(k)} - H^*\| \leq \frac{L}{1-L} \underbrace{\|H^{(k)} - H^{(k-1)}\|}_{\text{RMSE } r_k}$$

This inequality implies via a logical chain: as long as we observe the RMSE $r_k$ decaying at rate $L$, it is mathematically guaranteed that the true error $\|H^{(k)} - H^*\|$ also tends to 0 at the same rate $L$. If the operator $\Phi$ is a contraction mapping, then for any adjacent iteration steps:

$$\underbrace{\|H^{(k+1)} - H^{(k)}\|}_{r_{k+1}} = \|\Phi(H^{(k)}) - \Phi(H^{(k-1)})\| \leq L \cdot \underbrace{\|H^{(k)} - H^{(k-1)}\|}_{r_k}$$

Recursively, we obtain $r_k \leq L^k \cdot r_0$. This proves that: under a contraction mapping ($L < 1$), the RMSE $r_k$ must decay exponentially.

**Empirical Verification:** We tracked the RMSE $r_k$ for our pretrained Pythia-410M model. The results are shown in Figure 9 (Left).

- Log-Linear Fitting: To quantify the convergence rate and verify the exponential decay hypothesis ($r_k \propto L^k$), we linearize the relationship by taking the base-10 logarithm:

$$\log_{10}(r_k) \approx k \cdot \log_{10}(L) + C$$

This equation indicates that if the convergence is exponential, the RMSE curve should form a straight line on a semi-logarithmic scale, with the slope corresponding to $\log_{10}(L)$.
- Exponential Decay: In the active convergence phase (first $\sim 10$ iterations), the RMSE exhibits a strict linear trajectory on the semi-logarithmic scale ($R^2 > 0.95$), confirming the exponential decay law consistent with the contraction mapping theory.
- Effective Lipschitz Constant ($L$): The fitting yields $L \approx 0.345 < 1$.
- Practical Implication: $L \approx 0.345$ implies an extremely rapid convergence rate.
    - Reducing the error to 1% ($10^{-2}$) of the initial value requires only $\sim 4$ iterations.
    - Reducing the error to 0.1% ($10^{-3}$) requires only $\sim 6$ iterations.
- Precision Floor: The convergence stabilizes at the BFloat16 precision limit ($\sim 10^{-5}$), confirming the algorithm has reached the maximum precision allowed by the hardware.

### D.3. Iteration Also Converges to the Sequential Solution $H_{\text{seq}}$

"Sequential Inference" produces the standard autoregressive result $H_{\text{seq}}$, and the result of the parallel iteration $H^{(k)}$ also converges to $H_{\text{seq}}$.

**Verification:** To empirically verify this, we calculated the RMSE between the parallel state $H^{(k)}$ at iteration $k$ and the sequential ground truth $H_{\text{seq}}$. As shown in Figure 9 (Right), the distance $\|H^{(k)} - H_{\text{seq}}\|$ similarly exhibits exponential decay. Within $\sim 9$ iterations, the difference drops to the BFloat16 precision floor ($\sim 10^{-5}$).

### D.4. Equivalence: Parallel Solution $H^*$ Matches Sequential Result $H_{\text{seq}}$

Since the result of the parallel Jacobi iteration $H^{(k)}$ converges to both $H^*$ and $H_{\text{seq}}$, by the uniqueness of limits, we know that $H^* = H_{\text{seq}}$, proving the two are completely identical.

## E. Randomizing the Iteration Count

**Setup.** We compare two schemes for the number of Jacobi iterations used in the hidden-state update: (i) a *fixed iteration* scheme with $K = 2$; and (ii) a *randomized iteration* scheme where $K \sim \text{Unif}\{2, 3\}$. All other training settings are identical. At evaluation, we track the root mean squared change of hidden states $\text{RMSE}(k) = \|H^{(k)} - H^{(k-1)}\|$ over up to 30 iterations.

**Observation.** With *fixed* $K = 2$, the curve exhibits a pronounced spike at the first unseen step ($k = 3$), indicating overfitting to a specific computational depth and poor generalization beyond trained iterations. In contrast, *randomized* $K \in \{2, 3\}$ induces a smooth, near-exponential decay that persists well beyond the trained range, converging to the BFloat16 numerical floor.

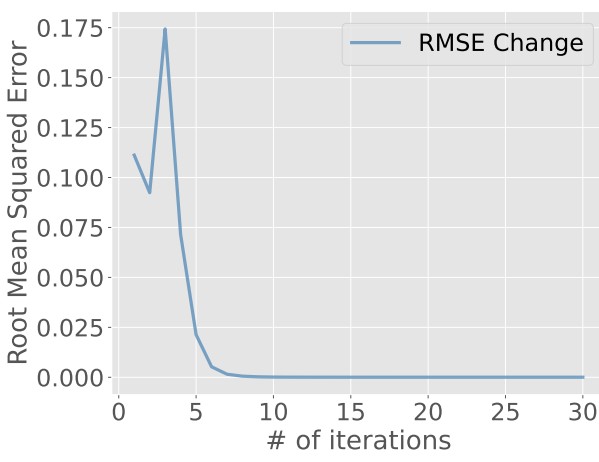 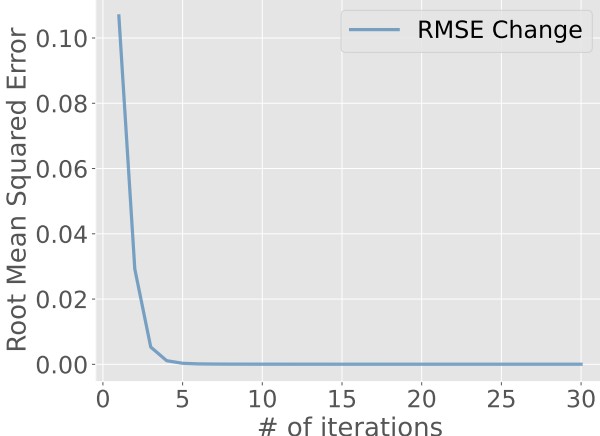

*Figure 12.* **Randomizing the iteration count prevents depth overfitting.** *Left:* Training with a fixed iteration count ($K = 2$) causes a large spike at the first unseen step ($k = 3$). *Right:* Randomized training ($K \in \{2, 3\}$) yields stable, near-exponential decay of $\|H^{(k)} - H^{(k-1)}\|$ and convergence to the numerical floor.

*Table 8.* Scaling the latent-thought chain length $K$ on LLaMA-410M (6B tokens). Language modeling is reported as perplexity (PPL; lower is better). Downstream tasks are reported as accuracy (higher is better), with the last column showing average accuracy.

| Model | Language Modeling ↓ | | Downstream Tasks (Acc↑) | | | | | | | | | Avg Acc |
|---|---|---|---|---|---|---|---|---|---|---|---|---|
| | Pile | Wikitext | Arc_e | Arc_c | Lamb-O | Wino | Piqa | Lamb-S | Hella | Sciq | Race | |
| 410M ($K = 0$) | 12.4 | 32.7 | 43.6 | 19.6 | 34.1 | 49.4 | 61.5 | 22.1 | 28.5 | 71.7 | 26.5 | 39.7 |
| 410M ($K = 1$) | 11.2 | 28.4 | 45.5 | 20.1 | 38.6 | 51.9 | 63.1 | 28.8 | 29.6 | 74.1 | 29.1 | 42.3 |
| 410M ($K = 3$) | 10.7 | 26.1 | 48.4 | 20.9 | 41.4 | 53.1 | 63.2 | 34.1 | 30.5 | 78.2 | 29.6 | 44.4 |
| 410M ($K = 5$) | 10.4 | 25.4 | 48.1 | 21.3 | 44.6 | 50.7 | 62.8 | 35.4 | 31.0 | 79.3 | 30.6 | 44.9 |

## F. Training Details

The primary computational cost comes from pretraining PonderLM-2-Pythia-1.4B on the 300B-token Pile dataset. This pretraining was conducted on a cluster of high-performance 64GB GPUs and required a total of 73,047 GPU hours.

## G. Scaling the Number of Latent Thoughts to a Larger 410M Model

To address concerns about the generalizability of our ablation findings, we further evaluate whether the benefits of increasing the number of latent thoughts persist at a larger scale. Specifically, we train a **LLaMA-410M** model on **6B tokens**. While our main experiments prioritize the 1.4B model due to computational constraints, these additional results confirm that the favorable scaling trend observed on smaller models transfers to a substantially larger backbone.

### G.1. Language Modeling and Downstream Tasks

We further extend the 410M experiments to include longer latent-thought chains, and evaluate both language modeling and downstream performance. As shown in Table 8, performance improves monotonically with larger $K$ across (i) language modeling perplexity and (ii) average downstream task accuracy.

Across Table 8, scaling up the number of latent thoughts remains beneficial on the **410M** model: (i) language modeling perplexity steadily decreases (e.g., Pile PPL drops from 11.2 at $K = 1$ to 10.4 at $K = 5$), and (ii) average downstream accuracy increases from 42.3% ($K = 1$) to 44.9% ($K = 5$). These results support the conclusion that **increasing the number of latent thoughts scales effectively to larger model sizes**, and the gains are not limited to smaller backbones.

*Table 9.* Ablation on LLaMA-410M trained on 6B tokens, evaluated by PPL on Pile validation. Numbers in parentheses denote absolute degradation relative to our full method.

| Method | PPL |
|---|---|
| Ours (**Hidden State + Jacobi + Interleaving**) | 11.197 |
| Replaced Hidden State w/ Pondering Embedding | 11.419 (+0.222) |
| w/o Jacobi Iteration | 11.622 (+0.425) |
| w/o Interleaving (Residual) | 11.748 (+0.551) |
| Vanilla baseline | 12.756 (+1.559) |

*Table 10.* Inference-time memory overhead and average downstream accuracy. Our method doubles KV-cache by design (one latent thought per token), yet remains more memory-efficient than competitive baselines that either require substantially more static parameters or incur larger KV-cache growth from multiple steps/loops/pauses.

| Method | Inference Memory Overhead (Source) | Avg Acc |
|---|---|---|
| Ours (LLaMA-1.4B) | $+1\times$ KV-cache (1 latent thought per token) | 51.9 |
| LLaMA-2.8B | $+1\times$ KV-cache (doubled layers) + 1.4B static parameters | 50.6 |
| Looped LLaMA-1.4B (4 loops) | $+3\times$ KV-cache (4 loop iterations per token) | 50.5 |
| Pause LLaMA-1.4B (3 pauses) | $+3\times$ KV-cache (3 pause tokens per token) | 50.4 |
| Pondering LLaMA-1.4B (3 steps) | $+3\times$ KV-cache (3 extra pondering steps) | 51.5 |

## H. Additional Ablation: Isolating Design Choices

To isolate the specific contributions of our design choices—**latent representation** (hidden state vs. pondering embedding), **Jacobi training**, and the **interleaving strategy**—we conduct an ablation study on a **LLaMA-410M** model trained on **6B tokens**. We report **perplexity (PPL)** on the **Pile validation set** (lower is better).

**Compared variants.** Starting from our full method, we consider:

- **Replaced Hidden State w/ Pondering Embedding.** We keep the overall architecture unchanged, but replace the latent representation with PonderLM's "probability-weighted sum of embeddings" rather than using the last hidden state.

- **w/o Jacobi Iteration.** We remove iterative parallel training by setting the number of Jacobi iterations to 0.

- **w/o Interleaving (Residual).** We use vertical scaling by adding the latent state to the current position embedding (a residual-style update, akin to PonderLM), instead of appending it as a new token (interleaving).

Table 9 shows that each component contributes to the final performance. In particular, removing **interleaving** or **Jacobi iteration** causes the largest regressions among the ablated components, indicating that (i) treating latent thoughts as *horizontally scaled* tokens and (ii) training them efficiently via Jacobi-style parallel iteration are both critical to achieving optimal perplexity.

## I. Inference Memory Overhead: KV-Cache Trade-offs

We acknowledge that our method increases inference-time memory due to the KV-cache: with *one latent thought per token*, the KV-cache size is effectively doubled compared to vanilla autoregressive decoding. Importantly, however, this overhead remains *more memory-efficient* than several competitive baselines we compare against (including doubled-parameter models in Section 4.3). These alternatives not only underperform relative to our method, but also incur substantially larger inference-memory footprints due to either (i) larger static parameter storage or (ii) multiple per-token forward passes that expand KV-cache proportionally.

Although our method increases KV-cache by $+1\times$, it avoids the substantially larger overheads incurred by baselines that rely on repeated per-token computation (e.g., loops/pauses/extra pondering steps), where KV-cache grows roughly linearly with the number of steps. Moreover, compared to doubling model size (e.g., LLaMA-2.8B), our approach achieves stronger

*Table 11.* Qualitative examples of latent-thought decoding. Stage 0 (Latent) often identifies the correct token as Top-1 with low probability; Stage 1 (Final) verifies and increases confidence.

| Case | GT Token | Stage 0 (Latent) | Stage 1 (Final) |
|---|---|---|---|
| Arithmetic: 8+6+1= | 15 | Prob: 0.07 (Top-1) | Prob: 0.48 (Top-1) |
| Retrieval: The capital of the Netherlands is | Amsterdam | Prob: 0.06 (Top-1) | Prob: 0.20 (Top-1) |
| Entity: The IATA code of United Airlines is | UA | Prob: 0.04 (Top-1) | Prob: 0.76 (Top-1) |

*Table 12.* Quantitative analysis on LAMBADA. Stage 0 has lower accuracy but exhibits substantial top-1 overlap with Stage 1. For overlap tokens, the mean probability increases significantly from Stage 0 to Stage 1.

| Horizontal Step | Accuracy | Overlap Rate | Mean Prob (Overlap Tokens) |
|---|---|---|---|
| Stage 0 (Latent) | 36.19% | – | 0.1522 |
| Stage 1 (Final) | 71.09% | 50.46% | 0.5826 |

downstream performance while avoiding the additional static parameter memory associated with a larger backbone. Overall, this provides a favorable trade-off between inference memory and task performance.

## J. Interpretability of Latent Thoughts via Intermediate Decoding

Based on our continually pretrained **LLaMA-3-3B** model (Section 4.4), we analyze what the proposed *latent thoughts* encode by probing intermediate decoding results. We compare two stages of generation:

- **Stage 0 (Latent).** We directly decode the latent thought (i.e., the hidden state) using the language model head.

- **Stage 1 (Final).** We feed the latent thought back into the model to produce the final prediction.

Our observations suggest that the latent thought acts as a *preliminary hypothesis generator*: it often surfaces the correct candidate with high uncertainty, while the subsequent step serves as a verification mechanism that substantially increases confidence.

### J.1. Qualitative Case Studies

Table 11 illustrates representative examples. Across diverse tasks (arithmetic, retrieval, and entity knowledge), Stage 0 frequently places the correct token as Top-1, but assigns it a low probability. After feeding the latent thought back into the model, Stage 1 increases the probability of the same token and yields a higher-confidence prediction.

### J.2. Quantitative Analysis on LAMBADA

We further quantify this phenomenon on the **LAMBADA** dataset. As shown in Table 12, directly decoding the latent thought (Stage 0) yields lower accuracy, yet it shares substantial top-1 overlap with the final output (Stage 1). Importantly, for tokens where Stage 0 and Stage 1 agree (overlap tokens), the mean probability assigned by Stage 0 is low and increases sharply after the additional computational step.

These results indicate that latent thoughts enable the model to *prepare* an answer in continuous space: Stage 0 often produces a weak but informative hypothesis (frequently aligned with the final token), and Stage 1 refines it into a high-confidence decision through an additional verification step.

## K. Peak Training Memory Profiling

A natural concern about our parallel training procedure is its memory footprint: the interleaved token–thought sequence doubles the effective context length, and each Jacobi iteration must store intermediate activations. We profile peak GPU memory under the same gradient-checkpointed setup used by both the vanilla continual-pretraining (CPT) baseline and PonderLM-2 CPT, and report the results in Table 13.

*Table 13.* Peak GPU memory during continual pretraining, measured under the same gradient-checkpointed setup for both baselines and PonderLM-2. Despite roughly $8\times$ training FLOPs, the relative memory overhead of PonderLM-2 stays at $\sim1.1\times$ at both scales, well below the overhead implied by the FLOPs multiplier.

| Model scale | Vanilla CPT | PonderLM-2 CPT | Overhead |
|---|---|---|---|
| LLaMA-3-3B | 70.9 GB | 77.1 GB | $1.09\times$ |
| LLaMA-3-8B | 110.8 GB | 124.8 GB | $1.13\times$ |

**Inference memory.** At inference, the additional memory cost is dominated by the KV-cache, which is analyzed separately in Appendix I. We summarize the training-time and inference-time pictures together: despite an $\sim8\times$ training-FLOPs overhead and a $2\times$ KV-cache, the *peak memory* footprint of PonderLM-2 remains close to that of vanilla training at both 3B and 8B, and remains more memory-efficient than several competitive baselines (Looped, Pause, PonderLM) that incur $\sim3\times$ KV-cache growth from multiple per-token forward passes.

## L. Attention Analysis of Latent Thought Queries

To better understand what computation latent thoughts actually perform, we analyze the attention patterns of latent queries in the final forward pass of our LLaMA-3-3B continually pretrained model. We measure attention across 64 sequences of length 2048, excluding the first two positions to remove attention-sink effects.

**Layer-wise attention allocation.** Figure 13(a) shows the proportion of attention mass that latent queries direct to original input tokens versus other latent thoughts at each layer. A clear layer-wise structure emerges: in both lower and upper layers, 60–80% of attention is directed toward the original tokens, while middle layers split attention roughly evenly between the two. This suggests that the original signal is preserved as an anchor throughout the network, while latent thoughts act as auxiliary context aggregators rather than replacing the original embeddings.

**Decomposition of original-token attention.** Figure 13(b) decomposes the attention mass directed to original tokens into two components: attention to *earlier* positions (broad context aggregation) and attention to the corresponding position's *own* original token (local refinement). Lower layers broadly aggregate context from earlier positions, while upper layers increasingly concentrate on the latent's own original token, consistent with a hierarchical "context-then-refine" pattern.

**Case studies at the last layer.** Figure 18 visualizes the attention weights of latent queries at Layer 27 on two representative sentences. In both cases, latent queries primarily attend to their corresponding and nearby original tokens, with a secondary attention pattern toward neighboring latent thoughts. This local, content-conditioned pattern is consistent with the quantitative trends in Figure 13 and supports the interpretation that latent thoughts serve as a structured refinement step rather than as parallel "thinking" tokens disconnected from the input.

## M. Learning Rate Ablation for Continual Pretraining

The main paper uses a peak learning rate of $3\mathrm{e}{-5}$ for the continual-pretraining (CPT) setup of Section 4.4. To verify that the observed loss dynamics of PonderLM-2 CPT are not specific to this choice, we additionally run the CPT at two flanking peak learning rates, $1\mathrm{e}{-5}$ and $5\mathrm{e}{-5}$ (a $5\times$ range around the main setting), on the same LLaMA-3-3B backbone and the same 5B-token SlimPajama stream.

Figure 14 shows the training loss curves under both ablation settings. The behavior is consistent across learning rates:

- **Smooth adaptation.** PonderLM-2 CPT exhibits a small, transient loss spike at the very start of CPT as the model accommodates the newly introduced latent states. This spike resolves quickly (within $\sim0.5$B tokens) and does *not* recur during the rest of training.
- **PonderLM-2 surpasses Vanilla CPT.** After the initial adaptation, PonderLM-2's loss drops below the vanilla CPT loss and remains consistently lower through the end of training. At LR $= 1\mathrm{e}{-5}$, PonderLM-2 reaches $\sim2.13$ vs. vanilla $\sim2.14$; at LR $= 5\mathrm{e}{-5}$, PonderLM-2 reaches $\sim2.14$ vs. vanilla $\sim2.165$.
- **Effect of learning rate.** The higher LR ($5\mathrm{e}{-5}$) leads to a faster crossover ($\sim1$B tokens) but a slightly higher absolute loss; the lower LR ($1\mathrm{e}{-5}$) gives a smoother curve and a lower final loss. Both flanking settings are stable throughout

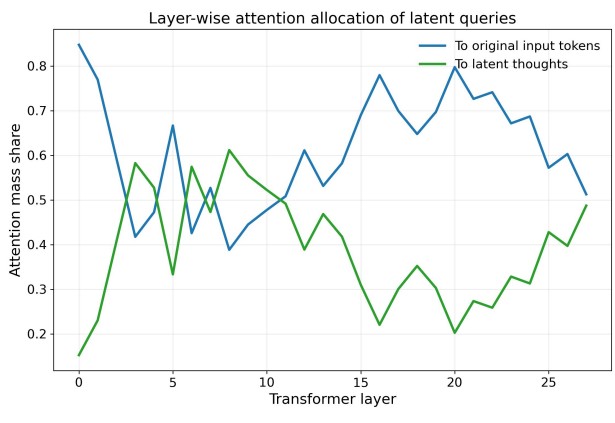 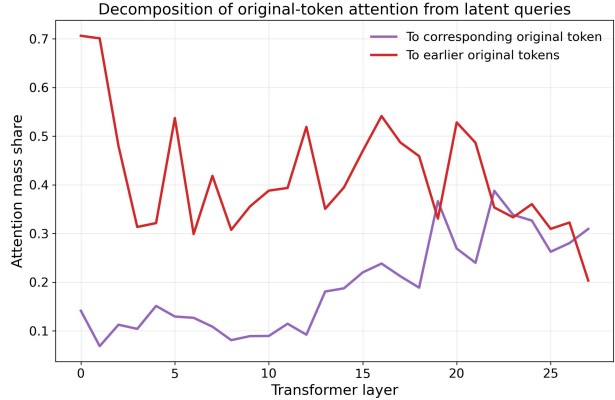

*(a)* Attention allocation of latent queries.     *(b)* Decomposition of original-token attention.

*Figure 13.* **Attention patterns of latent thought queries in the final forward pass** of the LLaMA-3-3B CPT model, measured across 64 sequences (2048 tokens each, excluding the first two positions to remove attention-sink effects). **(a)** Layer-wise attention mass from latent queries to original input tokens vs. other latent thoughts. Lower and upper layers direct 60–80% of attention to original tokens, while middle layers split attention roughly evenly. **(b)** Decomposition of the original-token attention: lower layers broadly attend to earlier positions (context aggregation), while upper layers increasingly focus on the corresponding position's own original token (local refinement).

training, which together with the main-paper run at $3\mathrm{e}{-}5$ covers a $5\times$ range of peak learning rates.

Together, these results indicate that the small initial loss spike is a generic adaptation phase to the new latent input format rather than a training-instability artifact, and that the loss-crossover behavior reported in the main text at $\mathrm{LR} = 3\mathrm{e}{-}5$ is robust across at least a $5\times$ range of peak learning rates.

## N. Jacobi Convergence on LLaMA-3-3B and Position-level Analysis

The convergence analysis in Section 4.6 is conducted on PonderLM-2-Pythia-410M. Here we replicate that analysis on the much larger LLaMA-3-3B continually pretrained model, and additionally provide a position-level breakdown of the convergence behavior.

**Global convergence rate at 3B.** Figure 15 reports the RMSE between the Jacobi iterate $H^{(k)}$ and the sequential fixed point $H_{\mathrm{seq}}$ on the LLaMA-3-3B CPT model. A log-linear fit yields an effective Lipschitz constant $L \approx 0.195$ ($R^2 \approx 0.979$), confirming exponential convergence at the 3B scale: the RMSE drops by over two orders of magnitude within $\sim$6 iterations and plateaus near machine precision thereafter. This contraction rate is comparable to the one reported on Pythia-410M in Figure 9, indicating that the rapid convergence observed at smaller scale carries over to a much larger backbone.

**Position-level convergence front.** Figure 16 dissects the convergence by token position. Panel (A) plots the per-position RMSE between $H^{(k)}$ and $H_{\mathrm{seq}}$ as a heatmap, with refinement iteration on the $y$-axis and latent position on the $x$-axis. A red dashed line marks the exact causal convergence boundary: positions to its left have already converged to machine precision. The shape of the front is consistent with the autoregressive causality argument in Appendix C: earlier positions converge first, and convergence propagates left-to-right with iteration count. Panel (B) tracks individual positions: position 1 converges after a single iteration, position 2 after two, while position 20 requires $\sim$7 iterations to reach machine precision ($< 10^{-7}$). This left-to-right front directly visualizes the empirical consistency between parallel Jacobi training and sequential autoregressive inference.

## O. Compute-Matched CoT Comparison

Because one PonderLM-2 inference sample costs roughly $2\times$ the inference FLOPs of a Standard CPT sample (one extra forward pass per token), a per-$N$ comparison of test-time scaling (Figure 8) is not strictly compute-matched. To control for this, we additionally compare under matched *inference FLOPs* on GSM8K with CoT prompting, using the same LLaMA-3-3B continual-pretraining models from Section 4.4.

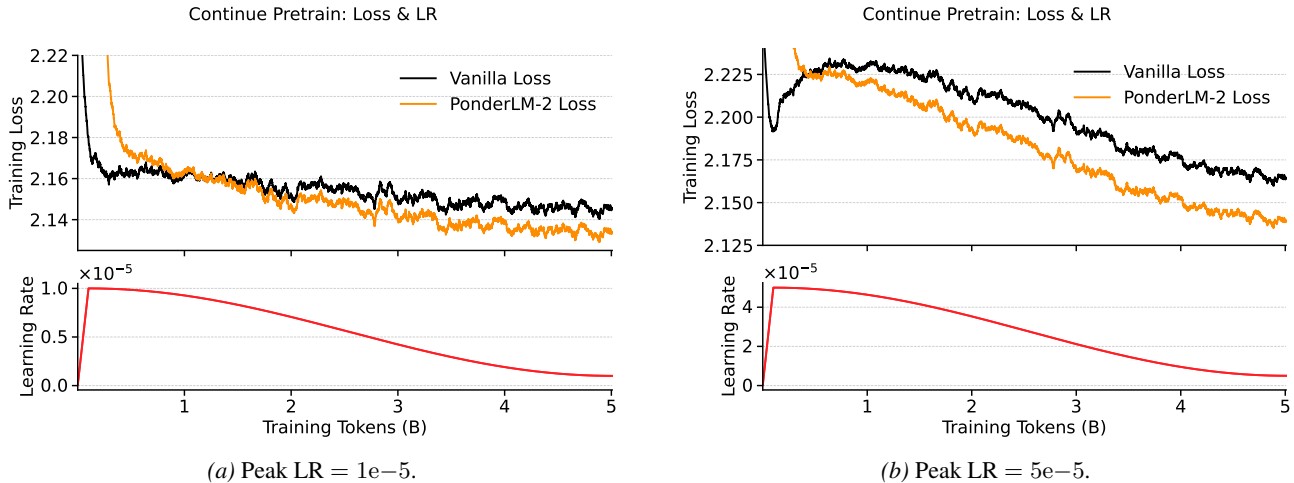

*(a)* Peak LR $= 1\mathrm{e}{-}5$.  *(b)* Peak LR $= 5\mathrm{e}{-}5$.

*Figure 14.* **Learning rate ablation for continual pretraining on LLaMA-3-3B (5B SlimPajama tokens).** Training loss of Vanilla CPT and PonderLM-2 CPT under two peak learning rates. PonderLM-2 exhibits a small, transient loss spike at the start as it accommodates the newly introduced latent states, then quickly recovers and surpasses Vanilla CPT. Higher LR ($5\mathrm{e}{-}5$) leads to a faster crossover but slightly higher final loss; lower LR ($1\mathrm{e}{-}5$) provides smoother convergence and a lower final loss. Both settings are stable throughout training.

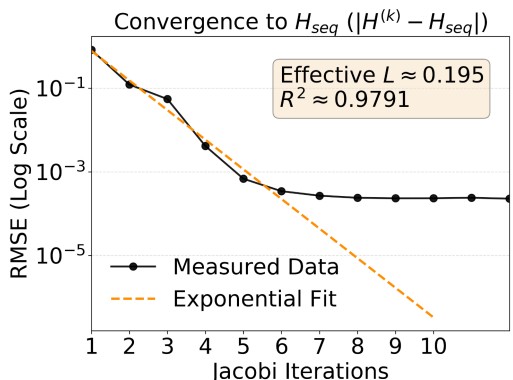

*Figure 15.* **Jacobi iteration convergence at the 3B scale.** RMSE between the Jacobi iterate $H^{(k)}$ and the sequential fixed point $H_{\text{seq}}$ on the LLaMA-3-3B CPT model, on a log scale. The effective Lipschitz constant is $L \approx 0.195$ ($R^2 \approx 0.979$): RMSE drops by more than two orders of magnitude within 6 iterations and plateaus near machine precision thereafter, mirroring the trend observed in Figure 9.

**Protocol.** For each compute budget, we pair $N$ PonderLM-2 samples against $2N$ Standard CPT samples, so that both methods consume the same number of inference FLOPs. We sweep $N$ over a wide range and report two aggregation strategies in Figure 17: majority voting (left) and best-of-$N$ (right). All other generation hyperparameters (temperature, prompt template, maximum new tokens) are held fixed across the two methods.

**Findings.** Under this fair compute-matched setting, PonderLM-2 remains *consistently better* on majority voting at all sampling budgets, and remains *comparable* on best-of-$N$. This rules out the possibility that the test-time scaling gains in Figure 8 are driven solely by the $2\times$ per-sample compute, and confirms that the underlying base model produced by PonderLM-2 CPT is genuinely stronger at the same inference budget. The strongest separation occurs under majority voting, suggesting that PonderLM-2 CPT produces samples that are clustered and consistently correct, making aggregation particularly effective.

## P. Long-Context Evaluation on RULER

Because PonderLM-2 interleaves a latent thought after every input token, a concern is whether the doubled sequence length harms long-context behavior. We note first that the usable context window is *not* reduced by our method: latent thoughts share the same positional encoding as their corresponding input tokens, each token still attends within its original window.

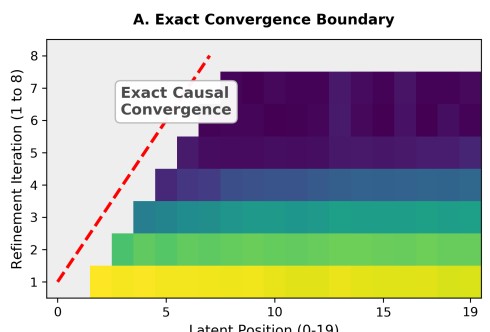
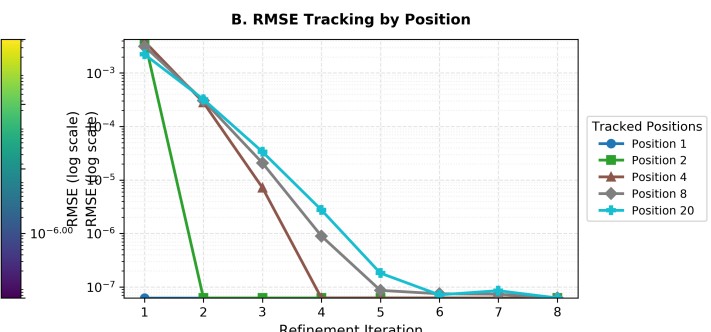

*Figure 16.* **Position-level convergence of Jacobi iteration on LLaMA-3-3B CPT. (A)** Heatmap of per-position RMSE between the Jacobi iterate $H^{(k)}$ and the sequential fixed point $H_{\text{seq}}$ across refinement iterations ($y$-axis) and latent positions ($x$-axis). The red dashed line marks the exact causal convergence boundary: positions to its left have already converged to machine precision. Earlier positions converge first, forming a clear left-to-right front consistent with the autoregressive causality argument. **(B)** RMSE trajectories for selected positions. Position 1 converges after 1 iteration, position 2 after 2, while position 20 requires $\sim$7 iterations. All positions ultimately reach machine precision ($< 10^{-7}$), empirically confirming both rapid and exact convergence.

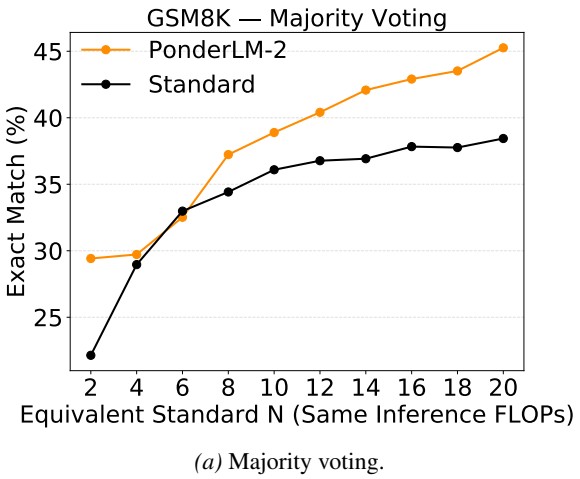
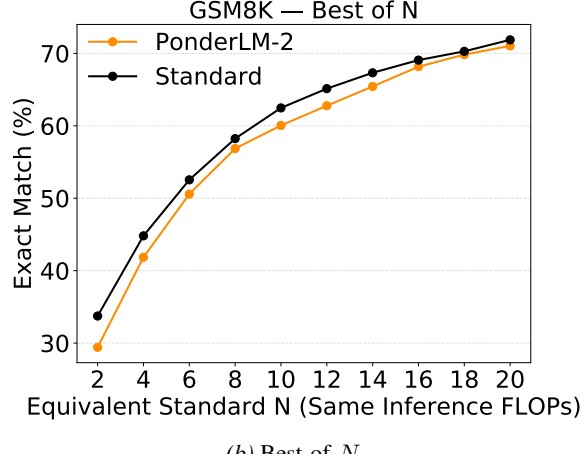

| (a) Majority voting. | (b) Best-of-$N$. |
| --- | --- |

*Figure 17.* **Compute-matched test-time scaling on GSM8K with CoT prompting.** Comparison between PonderLM-2 CPT and Standard CPT using the LLaMA-3-3B continual-pretraining models from Section 4.4. Because one PonderLM-2 sample costs $\sim2\times$ the inference FLOPs of a Standard CPT sample, we match $N$ *PonderLM-2 samples* against $2N$ *Standard CPT samples*. **Left:** majority voting. **Right:** best-of-$N$. Under this compute-matched protocol, PonderLM-2 is consistently better on majority voting and comparable on best-of-$N$, showing that the test-time scaling gains in Figure 8 persist after accounting for the per-sample FLOPs difference.

We directly evaluate this on RULER, a standard synthetic long-context benchmark, using the LLaMA-3-3B CPT models from Section 4.4. We report average RULER scores at two context lengths, 4K and 8K (Table 14).

PonderLM-2 CPT consistently outperforms the matched vanilla baseline at both context lengths (74.44 vs. 54.08 at 4K; 61.43 vs. 40.40 at 8K). Crucially, its drop from 4K to 8K ($-13.02$) is comparable to and slightly smaller than that of vanilla CPT ($-13.68$), so the added latent tokens do not cause a disproportionate long-context regression. Efficient-attention techniques such as sliding-window attention and GQA are orthogonal to our approach and can be combined with it to further mitigate any long-sequence overhead.

## Q. Coding, Math, and Reasoning Benchmarks for LLaMA-3-3B CPT

The general-domain downstream evaluation in Table 4 is dominated by commonsense and reading-comprehension tasks. To better characterize where PonderLM-2 CPT helps on a stronger backbone, we additionally evaluate the LLaMA-3-3B CPT models on more targeted coding, math, and reasoning benchmarks (Table 15).

PonderLM-2 CPT improves over the vanilla baseline on every benchmark, with the largest gains on coding (MBPP $+8.93$,

*Table 14.* Long-context evaluation on RULER (average score across subtasks) using the LLaMA-3-3B CPT models. PonderLM-2 CPT outperforms vanilla CPT at both 4K and 8K context lengths, and its degradation from 4K to 8K is comparable to (and slightly smaller than) the vanilla baseline, indicating that interleaving latent thoughts does not disproportionately degrade long-context behavior.

| Model | RULER @ 4K | RULER @ 8K | $\Delta$ (4K $\rightarrow$ 8K) |
|---|---|---|---|
| Vanilla CPT (LLaMA-3-3B) | 54.08 | 40.40 | $-13.68$ |
| PonderLM-2 CPT (LLaMA-3-3B) | **74.44** | **61.43** | $-13.01$ |

*Table 15.* Coding, math, and reasoning benchmarks for the LLaMA-3-3B continual-pretraining models from Section 4.4. PonderLM-2 CPT yields gains across coding (MBPP, HumanEval), math (MATH500, AMC23), and knowledge reasoning (OpenBookQA) benchmarks.

| Benchmark | Vanilla CPT | PonderLM-2 CPT | $\Delta$ |
|---|---|---|---|
| MBPP (3-shot) | 27.72 | **36.65** | $+8.93$ |
| HumanEval (0-shot) | 17.68 | **22.56** | $+4.88$ |
| MATH500 (pass@16) | 21.80 | **26.40** | $+4.60$ |
| AMC23 (pass@16) | 10.00 | **15.00** | $+5.00$ |
| OpenBookQA (0-shot) | 25.00 | **31.20** | $+6.20$ |
| OpenBookQA (5-shot) | 31.60 | **35.20** | $+3.60$ |

HumanEval +4.88) and math (MATH500 +4.60, AMC23 +5.00), and a consistent improvement on knowledge-oriented reasoning (OpenBookQA +6.20 at 0-shot). Because both models start from the same LLaMA-3-3B backbone and consume the same 5B SlimPajama tokens, these gains can be attributed to the added per-token latent computation rather than to a richer training mixture. Combined with the language-modeling and downstream-task gains reported in Section 4.4, this indicates that the benefits of PonderLM-2 generalize to reasoning-intensive domains and not just to commonsense tasks.

# R. Factual and Knowledge-Intensive Benchmarks (Pythia Scale)

A natural question for any horizontally-scaled method is whether the latent computation helps the model store and recall *factual* knowledge, or whether the gains are confined to local language modeling. To probe this directly, we evaluate PonderLM-2-Pythia-1.4B and the official Pythia-2.8B (which has roughly 2$\times$ the parameters) on several factual / knowledge-intensive benchmarks (Table 16).

Despite having only half the parameters, PonderLM-2-Pythia-1.4B *outperforms* Pythia-2.8B on every evaluated factual benchmark, with the largest relative gain on TriviaQA (+5.42 absolute, a ~55% relative improvement). These results suggest that the added latent computation does not merely smooth the language-modeling distribution but actually helps the model utilize its (smaller) parameter store more effectively for factual recall and knowledge-intensive question answering. Combined with the coding/math gains in Appendix Q, this indicates that the parameter-efficiency advantage of PonderLM-2 extends beyond the commonsense benchmarks of the main paper.

*Table 16.* Factual and knowledge-intensive benchmarks comparing PonderLM-2-Pythia-1.4B with the larger official Pythia-2.8B (both pretrained on 300B tokens from the Pile). Despite having only half the parameters, PonderLM-2-Pythia-1.4B outperforms Pythia-2.8B on every benchmark, suggesting that latent thoughts help the model exploit its parameters more effectively for factual recall.

| Benchmark | Pythia-2.8B | PonderLM-2-Pythia-1.4B |
|---|---|---|
| TriviaQA (Acc) | 9.83 | **15.25** |
| NQ-Open (Acc) | 3.91 | **4.18** |
| TruthfulQA (BLEU / ROUGE) | 26.07 / 23.26 | **28.40 / 26.86** |
| WebQuestions (Acc) | 2.81 | **3.44** |

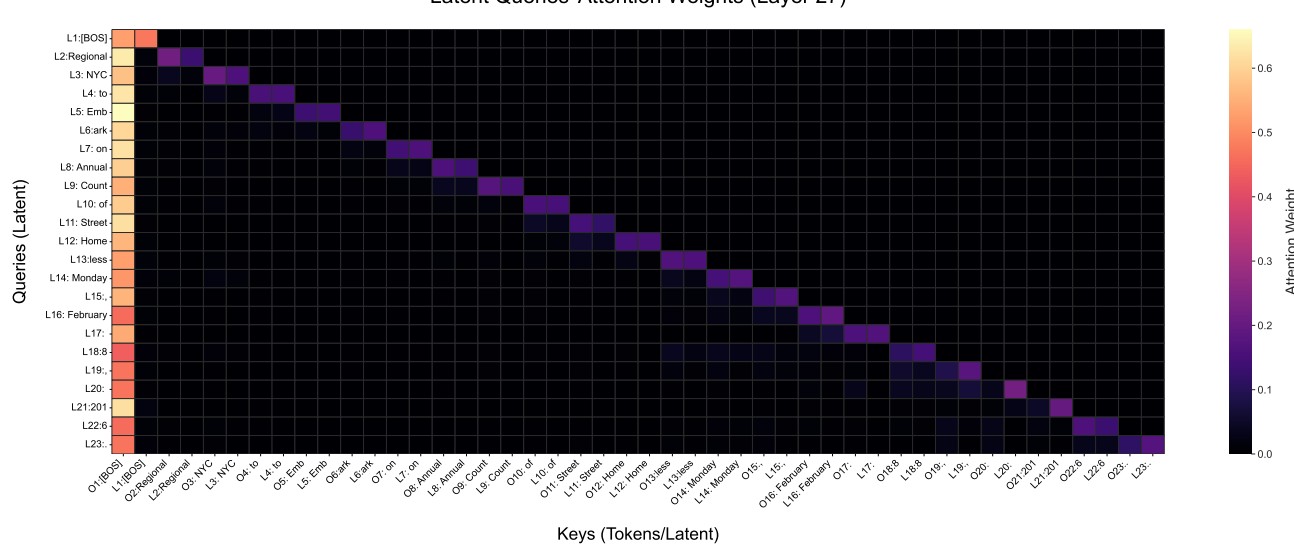

*(a)* Case 1: "Regional NYC to Embark on Annual Count of Street Homeless Monday, February 8, 2016."

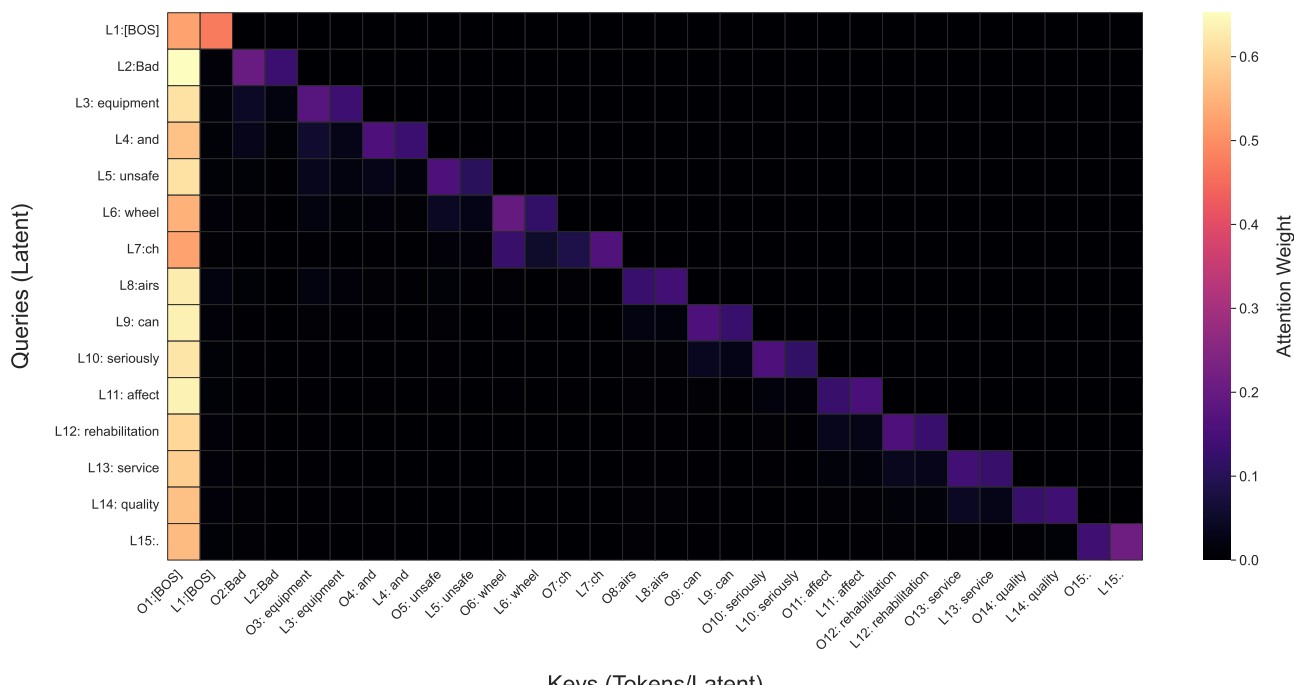

*(b)* Case 2: "Bad equipment and unsafe wheelchairs can seriously affect rehabilitation service quality."

*Figure 18.* **Case studies of latent query attention at the last layer (Layer 27) of the LLaMA-3-3B CPT model.** Each heatmap shows the attention weights from latent queries (rows, prefixed L) to all keys (columns: original tokens prefixed O, latent tokens prefixed L). In both examples, latent queries primarily attend to their corresponding and nearby original tokens, with a weaker secondary band on neighboring latent thoughts.

