# OpenReview forum: "PonderLM-2: Pretraining LLM with Latent Thoughts in Continuous Space"
_ICML.cc/2026/Conference — ICML 2026 spotlight_

### Official Review · Reviewer_q5m2 · 2026-03-08

**Soundness:** 4
**Presentation:** 4
**Significance:** 4
**Originality:** 4
**Overall Recommendation:** 6
**Confidence:** 4

**Summary:**

Recurrent transformer or soft thinking proposes to improve the performance by using the last hidden state as the next input. The paper proposes PonderLM-2, which inputs both discrete tokens and the last hidden state in each step. One major limitation of the recurrent transformer is its training difficulty. Its recurrent nature makes it hard to be trained in parallel. The paper proposes a new parallel training method called Jacobi Iteration. The method could be seen in Figure 3. The main idea is that we iteratively approximate the last hidden state by forward passes. The experiments show that the resulting Pythia variants are much better than Pythia with the same size. In some tasks, it is even better than the Pythia with double sizes.

**Compliance With Llm Reviewing Policy:**

Affirmed.

**Final Justification:**

The rebuttal addresses all my concerns. I think this paper is a good candidate for the (best paper) award due to its potential impact.

**Key Questions For Authors:**

1. Recurrent transformer and continuous thinking were originally designed for logical/math/reasoning problems. Their original motivation was that the number of layers in the transformer might not be enough compared to the required logical steps. Your work shows that a recurrent transformer could be trained efficiently and greatly improve the questions in lots of domains other than math and coding. Why do you think the math/coding/STEM in MT-bench do not improve significantly? Is it because Pythia 1.4B or 0.4B too small to answer correctly for most of math/coding questions?

2. In the standard transformer, the computation at each step is driven by the input token. When we add the hidden state as another input, does the input token become less important? Would the proposed approach encourage some hidden thinking of LLMs and make the thinking thread or explanation less faithful?

3. Could you compare the memory usage during the training time?

4. Could you also put the performances of a larger standard LM in Figure 8? The current comparison does not seem to be totally fair due to its double inference cost.

4. Could you compare PonderLM-2-Pythia-1.4B with Pythia-2.8B on some factuality benchmarks and see which one can recall more facts / answer knowledge-intensive questions?

**Limitations:**

yes

**Strengths And Weaknesses:**

Strength of Soundness

The methods are simple and effective.
They conduct lots of experiments to verify the effectiveness of the proposed approach.
The improvements are very significant.

----------------

Strength of Presentation

The presentation is very clear. All figures and tables are designed well.

----------------

Strength and Weakness of Significance

Jacobi iteration is an elegant approach that overcomes the biggest problem of the recurrent transformer studies. It is really impressive that it can achieve much better performance given the same FLOPs in Figure 1.

Nevertheless, the proposed approach is not universally better than the standard transformer. First, I guess the gradients will need to flow through multiple forward passes in Figure 3, which means the training memory usage might be much larger than the standard transformer. Second, it needs to double the sequence length during the inference time, so its inference FLOPs are similar to the transformer with a double size (even though it needs smaller memory to save the parameters).

The limitations create some concerns. First, the standard model with double sizes might be able to store more factual knowledge in its parameters. Second, the attention complexity is n^2, so the standard model might be more memory efficient for long-context tasks. Third, the high training memory might prevent us from training a larger model.

Another source of the concerns is its training approximation method. Training stability is always a big issue as the LLM gets larger. It is possible that the error in this approximation will make training larger LMs unstable. It is also possible that the approximation error could reduce the accuracy of some downstream applications.

In short, if we can still have similar improvement as the model gets larger, it could become a mainstream method for LLM training. Look forward to the future work of PonderLM-2!

----------------

Strength of Originality

This paper does not create a research direction that most people are unaware of. Nevertheless, as far as I know, the parallel training procedure via Jacobi iteration is a new method to solve an important engineering problem.

---

> ### Author Rebuttal · Authors · 2026-03-31
>
> ---
>
> > [W1] Training memory & inference FLOPs.
>
> Training memory. Under the same gradient-checkpointed setup, peak memory overhead is modest: 3B vanilla CPT 70.9 GB vs. PonderLM-2 77.1 GB (\~1.09×); 8B vanilla CPT 110.8 GB vs. PonderLM-2 124.8 GB (~1.13×).
>
> Inference FLOPs. At matched 2× inference FLOPs, our method outperforms all baselines including the doubled-parameter LLaMA-2.8B: 51.9 vs. 50.6 avg 5-shot accuracy, with lower perplexity on all benchmarks.
>
> ---
>
> > [W2] Concerns:  (1)  less memory-efficient ; (2) high training memory .
>
>
> 1. **Long-context memory**: Our +1× KV-cache overhead (Table 10, Appendix I) is smaller than comparable multi-step baselines (Looped/Pause/Pondering incur ~+3× KV-cache). A 2.8B model also incurs +1× KV-cache plus substantially more parameter storage. Efficient techniques (GQA, KV-cache quantization) are orthogonal and directly applicable.
>
> 2. **Training memory at scale**: With gradient checkpointing, overhead is only ~1.1×. Our memory profiling on 8B (110.8 → 124.8 GB) confirms no practical barrier at this scale.
>
> ---
>
> > [W3] Training stability
>
> Training is stable across all scales tested (410M to 3B), with smooth loss curves and no instability. Jacobi iteration converges with similar as model size grows.
>
> ---
>
>
> > [Q1]math/coding/STEM
>
> We believe this likely reflects a combination of model scale and pretraining data quality. Pythia-0.4B/1.4B are relatively small, and The Pile contains much less high-quality math/code data than modern pretraining mixtures, which likely limits gains on MT-Bench math/coding/STEM categories.
>
> To better distinguish whether this limitation is method-driven or data/model-driven, we evaluated the LLaMA-3-3B continual-pretraining models from Section 4.4 (Vanilla CPT and PonderLM-2 CPT, both trained on SlimPajama) on more targeted coding, math, and reasoning benchmarks:
>
> | Benchmark | Vanilla CPT | PonderLM-2 CPT | Δ |
> |-----------|------------|----------------|---|
> | MBPP (3-shot) | 27.72 | **36.65** | +8.93 |
> | HumanEval (0-shot) | 17.68 | **22.56** | +4.88 |
> | MATH500 (pass@16) | 21.80 | **26.40** | +4.60 |
> | AMC23 (pass@16) | 10.00 | **15.00** | +5.00 |
> | OpenBookQA (0-shot) | 25.0 | **31.2** | +6.20 |
> | OpenBookQA (5-shot) | 31.6 | **35.2** | +3.60 |
>
> PonderLM-2 CPT shows clear improvements on coding (MBPP +8.93, HumanEval +4.88), math (MATH500 +4.60, AMC23 +5.00), and reasoning/STEM (OpenBookQA +6.20 / +3.60). Since LLaMA-3-3B was originally pretrained on a much richer data mixture including code and math, these results suggest that the weak MT-Bench gains in the Pythia setting are more likely due to data/model limitations than an inherent weakness of the method on these domains.
>
> ---
>
>
> > [Q2] attention analysis
>
> Our quantitative attention analysis (see R1-Q2) shows that original input tokens remain important. Latent thoughts‘ queries direct 60–80% of their attention mass to original input tokens in lower and upper layers, with the corresponding position's original token receiving increasing attention in upper layers. The latent thought does not replace the original signal — it acts as an auxiliary context aggregator alongside it.
>
> Regarding faithfulness: Appendix J shows that the latent thought (Stage 0) and final prediction (Stage 1) frequently agree on the top-1 token (50.46% overlap on LAMBADA), suggesting the latent process is aligned with the explicit prediction.
>
> ---
>
>
> > [Q3] Memory usage during training?
>
> See R1-W1. With gradient checkpointing (used by both methods), the overhead is ~1.1×.
>
> ---
>
> > [Q4]  fairer comparison.
>
> To address this, we compare under matched inference FLOPs using the LLaMA-3-3B continual-pretraining models: PonderLM-2 with \(N\) samples versus Standard CPT with \(2N\) samples, under the same CoT prompting and evaluation protocol. This is also the cleanest control for Figure 8, since in the Section 4.4 LLaMA setup there is no Standard counterpart that is close to \(2\times\) larger; the next available scale would substantially change both model size and compute, rather than isolating the effect of reasoning-time budget. Under this fairer compute-matched comparison, PonderLM-2 remains consistently better on majority voting and comparable on best-of-\(N\) on GSM8K (https://anonymous.4open.science/status/ponder2_photos-7B41). We will add these matched-compute Standard baselines to Figure 8 in the camera-ready version.
>
> ---
>
>
> > [Q5] factuality benchmarks.
>
> We evaluated PonderLM-2-Pythia-1.4B and Pythia-2.8B on several factual/knowledge-intensive benchmarks. Despite having half the parameters, PonderLM-2-Pythia-1.4B performs better on all evaluated tasks: TriviaQA (15.25 vs. 9.83), NQ-Open (4.18 vs. 3.91), TruthfulQA (28.40/26.86 vs. 26.07/23.26 in BLEU/ROUGE), and WebQuestions (3.44 vs. 2.81). These results suggest that latent thoughts can help the model utilize its parameters more effectively for factual recall and knowledge-intensive question answering. We will include these results in the camera-ready version.

---

> > ### Author Rebuttal · Reviewer_q5m2 · 2026-04-02
> >
> > Thanks for the rebuttal. I raise my score to 6. I have one more minor question out of curiosity. The overhead of gradient checkpointing is not high for a normal LLM. But I guess its overhead for PonderLM-2 is higher. Could you also provide the time overhead of gradient checkpointing for PonderLM-2 and the memory overhead of PonderLM-2 without using gradient checkpointing?

---

> > > ### Author Response · Authors · 2026-04-03
> > >
> > > Thank you for raising your score and for this insightful follow-up. We ran a separate controlled profiling experiment on LLaMA-3-3B under the same fixed setup, toggling gradient checkpointing (GC) on and off for both vanilla CPT and PonderLM-2 CPT:
> > >
> > > Time overhead of GC: PonderLM-2's GC slowdown ratio is **1.18×**, nearly identical to vanilla's **1.17×**. This confirms that GC does **not** disproportionately penalize PonderLM-2.
> > >
> > > Memory overhead without GC: PonderLM-2 uses **~2.02×** the memory of vanilla (expected, due to the Jacobi iteration). With GC enabled, this drops to only **~1.09×**, because GC substantially reduces the dominant activation storage from Jacobi iterations — making it particularly effective for our method. Since GC is widely adopted in large-scale LLM pretraining, the GC-enabled setting reflects the realistic training configuration.
> > >
> > > Thank you again for the constructive feedback, which has meaningfully improved the paper.

---

### Official Review · Reviewer_ngAf · 2026-03-10

**Soundness:** 4
**Presentation:** 4
**Significance:** 2
**Originality:** 3
**Overall Recommendation:** 5
**Confidence:** 4

**Summary:**

This paper introduces PonderLM-2, and its novel pretraining methodology. The main idea revolves around enabling autoregressive language models to generate intermediate latent embeddings, before committing to predicting an actual discrete token. In practice the way this is implemented during pre-training, is by utilizing a Jacobi iteration trick to break the sequential dependency that stems from recursively passing the predicted latent embedding into the next time step. This way, the pre-training process can to a large extent be parallelized as each forwardpass now requires only a small number of jacobi iterations, rather than a full rollout.

This methodology is ultimately compatible with the established autoregressive transformer architecture, allowing it to be easily applied to existing models. From various experiments, the authors demonstrate that this method results in increased performance, both on language modelling and downstream tasks. The tradeoff being an increased cost (and complexity) during pre-training.

**Compliance With Llm Reviewing Policy:**

Affirmed.

**Final Justification:**

Given the added experiments regarding the methods scaling, I consider this paper worthy of acceptance.

**Key Questions For Authors:**

Do you have any intuitions or evidence of how this method might scale?
Meaning, will the gains demonstrated by this method keep being significant, or will they eventually become negligible. I would love to hear your thoughts on this.

**Limitations:**

See strengths and weaknesses

**Strengths And Weaknesses:**

The methodology is clearly described and intuitive, with the Figure 2 and 3 being very pedagogical. The idea of using Jacobi Iteration as a means to break the casual dependency is indeed very neat. Furthermore, the method's ability to work fairly seamlessly with established transformer architectures is itself a strength.

The results from the extensive experiments are convincing, and tell a clear and promising story. Showing both good performance on language modeling and downstream tasks.
In particular I like the results brought about by section 4.5 regarding CoT as this seems like the most important baseline to this method in my opinion. Indeed, I would have wished for a deeper and more extensive comparison to regular CoT. But there is only so much space in 8 pages, and the provided results are promising.

That being said, I would advise you to be a bit more clear regarding the explanation in section 4.1.1. It is not clear to me if the results presented in Figure 1 are from training data or test data. Please clarify this both in the text and the figure caption.

The main weakness I see with the paper is its currently limited impact and relevance to the field. While the paper clearly demonstrates a promising direction, the step towards that direction is not gigantic. Something that becomes limiting, as it provides a clear trade-off between pre-training cost and inference results.

For this paper to get a stronger verdict than a “Weak Accept”, it would in my opinion have to more clearly demonstrate how this method scales, or show that the quality it produces is immediately useful for people. In a sense, I would like to see a more convincing argument that motivates the large increase in pre-training cost.

---

> ### Author Rebuttal · Authors · 2026-03-31
>
> Thank you for recognizing the clarity of our method, the neat Jacobi-iteration-based design, the compatibility with standard Transformer architectures, and the convincing empirical results, especially the promising CoT-related findings. Below, we provide point-by-point responses to your comments.
>
> ---
>
> > [W1] Clarify whether Figure 1 uses training or test data.
>
> Figure 1 shows results on **held-out test data**. We will clarify this in both the text and figure caption in the camera-ready version.
>
> ---
>
> > [W2] Limited impact and scaling; need more convincing argument for the pre-training cost trade-off.
>
> We have strengthened the scaling argument:
>
> 1. **Scaling curves** (Figure 10, Appendix B): from 405M to 1.4B on both GPT-2 and LLaMA, PonderLM-2's advantage is consistent and **does not narrow** with scale.
> 2. **LLaMA-3-3B CPT** (Section 4.4): significant improvements across language modeling, downstream tasks, and test-time scaling (CoT, BoN, majority voting), demonstrating that PonderLM-2 provides meaningful gains at the 3B scale.
> 3. **Practical cost**: Continual pretraining from existing checkpoints is substantially cheaper than training from scratch and already yields clear gains. This is the practical deployment path for enhancing existing foundation models.
>
> ---
>
> > [Q1] Do you have any intuitions or evidence of how this method might scale?
>
> We believe a particularly practical scaling path is through continual pretraining on strong off-the-shelf models, which requires only a small additional compute budget rather than full pretraining from scratch. In Section 4.4, we show that continuing pretraining LLaMA-3-3B for only 5B SlimPajama tokens already yields significant gains in language modeling, downstream tasks, and test-time scaling. On reasoning-intensive benchmarks, PonderLM-2 CPT also substantially outperforms vanilla CPT (e.g., MBPP +8.93, MATH500 +4.60, OpenBookQA +6.20). Moreover, PonderLM-2 CPT benefits more from CoT than vanilla CPT (+20.31 vs. +14.25 on GSM8K), suggesting that the latent computation continues to compound with additional inference-time scaling rather than becoming redundant. Overall, these results suggest that the method scales not only through larger pretraining runs, but also through lightweight continual pretraining, making it a practical way to upgrade existing foundation models with relatively modest extra compute.

---

> > ### Author Rebuttal · Reviewer_ngAf · 2026-04-03
> >
> > Thank you for further extending the scaling experiments (at least slightly). Given the new experiments I raise my score to Accept.

---

> > > ### Author Response · Authors · 2026-04-06
> > >
> > > We sincerely thank the reviewer for acknowledging that our responses have adequately addressed the concerns and for increasing the score. We also appreciate the reviewer’s comments, which have meaningfully helped us improve the paper.

---

### Official Review · Reviewer_KvMB · 2026-03-11

**Soundness:** 3
**Presentation:** 3
**Significance:** 3
**Originality:** 4
**Overall Recommendation:** 4
**Confidence:** 4

**Summary:**

This paper introduces a new pre-training methodology for language models with latent thoughts (PonderLM-2), which enhances generation quality by scaling generation steps at test time.
Specifically, the main idea of PonderLM-2 is to condition the generation of every token on an additional latent thought tokens, thereby doubling the number of generated tokens and increasing inference as well as training compute.
The pre-training method PonderLM-2 comprises 3 steps, where first, the initial hidden states (latent thought tokens) are estimated with a forward pass, and second, interleaved with the original embeddings, iteratively refined through additional parallel forward passes. Finally, the loss is computed for predicting the output tokens from the latent thought tokens.
Experiments comprising pre-training models with 1.4B parameters and 300B tokens as well as fine tuning existing off-the-shelf LLMs demonstrate that PonderLM-2 consistently improves the models language modeling performance over their default counterparts.

**Compliance With Llm Reviewing Policy:**

Affirmed.

**Final Justification:**

As stated below the authors rebuttal addressed all my questions. I maintain my acceptance score.

**Key Questions For Authors:**

1) Is the convergence rate always the same? Does it depend on model size and/or training data?
2) Why is the method called Jacobi iteration? Jacobi iteration is an algorithm for solving linear systems of equations. It seems that the main similarity is the iterative nature of the proposed method in the paper and the original Jacobi iteration.
3) Are the gradients backpropagated through the iterative hidden state (latent thought token) refinements or are they detached after the Kth iteration?
4) Figure 7: why limit the number of chained latent thoughts to 3? It seems that the loss could still improve with more latent thought tokens.
5) Did the authors observe that the hidden states converge from left to right or track RMSE of individual hidden state positions?


Comments:
- “Saturating scaling laws” is a very strong statement. The references lack support for a statement of this generality. A better phrasing would be: “Scaling laws predict diminishing returns…”
- The scaling plot in Figure 10 would be more conclusive if FLOPs were plotted on the x-axis.

**Limitations:**

yes

**Strengths And Weaknesses:**

Presentation:
- (Strong) PonderLM-2 is very well presented and the paper is well structured. PonderLM-2 is well placed in the recent literature and in addition a nice overview about related methods including their differences among each other are provided.
   - Table 1 is nice (adding references to each method would make it even more accessible)
- (Weak) For some figures it would help understandability if it was mentioned which model (size & training tokens) was used to produce the results. E.g. For Figure 9, one needs to dig deep into the Appendix (L.780).

Soundness:
- The experiments are extensive (300B tokens, 1.4B params), key hyperparameters and design choices of PonderLM-2 are ablated (number of Jacobi iterations, different position embeddings, number of latent thought tokens)
- There is a convergence analysis on the Jacobi iteration in the appendix and there is also empirical evidence (Figure 9) that shows that in the evaluated settings the Jacobi iteration converges.
  - However the empirical convergence analysis could be more tightly coupled to the theoretical argumentation:
It seems that the main argument for the convergence analysis is the autoregressive causality of transformers. Did the authors observe that the hidden states converge from left to right or track RMSE of individual hidden state positions?
- The claim that the PonderLM-2 model (1.4B params) is better than a two times larger standard Transformer model (2.8B params). However, this claim could be better supported by including accurate FLOP numbers across model scales. E.g. one could visualize the scaling curve in Figure 10 for training FLOPs and also compare the performance with the inference FLOPs.

Significance:

- A major problem with current LLMs is their computational cost for long sequences. PonderLM-2 adds additional intermediate tokens to each token produced. This significantly increases context lengths and therefore further increases the major bottleneck of current LLMs (the computational cost for long sequences).
  - For a fair comparison, one would need to compare this method in a reasoning setting, where the default LLM produces a reasoning trace and the PonderLM should produce a shorter trace (including thinking tokens)
  - Nevertheless, as this method is a pre-training method is a first step in this direction and shows strong results in the evaluated settings.
- Another axis of evaluation which is missing and would further improve the significance of the contribution, would be the inclusion of long-context evaluations e.g. RULER.
  - Since the PonderLM increases context length significantly, it would be interesting whether the additional latent thought tokens impact long context performance.
Again, as PonderLM-2 is a first step towards “pondering language models” with experiments of decent scale, I do not weigh this too strongly.

Originality:

- This paper introduces PonderLM-2, a novel pretraining method for LLMs. To the best of my knowledge the method is novel.

---

> ### Author Rebuttal · Authors · 2026-03-31
>
> > [W1] Figure Details.
>
> We will update all figure captions to explicitly state the model size and training token count. Figure 9 will state it uses PonderLM-2-Pythia-410M pretrained on 300B tokens from the Pile.
>
> ---
>
> > [W2] Table 1
>
> We will add references to each method in Table 1.
>
> ---
>
>
> > [W3] RMSE left to right.
>
> We have verified this on the LLaMA-3-3B CPT model with a **position-level convergence heatmap** (https://anonymous.4open.science/status/ponder2_photos-7B41). Panel A shows the exact convergence boundary: earlier positions converge to the sequential fixed point within 1–2 iterations, while later positions require proportionally more iterations, forming a clear left-to-right convergence front consistent with our theory. Panel B tracks RMSE for individual positions, confirming that Position 1 converges after 1 iteration, Position 2 after 2, while Position 20 requires ~7 iterations — all reaching machine precision.
>
>
>
> > [W4] Scaling Claim and Figure FLOPs.
>
> Table 3 already reports inference wall-clock throughput,shows that our 1.4B model outperforms the 2x-parameter LLaMA-2.8B under matched inference compute.  We agree that adding more explicit FLOPs information to the scaling curves and performance comparisons would further strengthen the claim, and we will include such annotations in Figure 10 and other relevant figures and tables.
>
> ---
>
>
> > [W5] Long-Sequence Cost
>
> At comparable wall-clock throughput (Table 3: 111.55 vs 221.19 tokens/s), our 1.4B model outperforms the 2x-parameter LLaMA-2.8B baseline (51.9 vs 49.8 avg accuracy) — achieving better quality with fewer parameters. Note that our method does not reduce the usable context window, as latent thought tokens share the same position IDs as their corresponding input tokens (Section 3.4). Furthermore, efficient attention techniques (sliding window, GQA) are orthogonal to our approach and can be  applied to mitigate long-sequence overhead.
>
> > [W6] Matched reasoning-time compute comparison
>
> Since generated reasoning traces are difficult to control to the same length across models, we instead compare under matched inference FLOPs using the LLaMA-3-3B continual-pretraining models. Concretely, since one PonderLM-2 sample costs roughly \(2\times\) a Standard CPT sample, we compare PonderLM-2 with \(N\) samples against Standard with \(2N\) samples, using the same CoT prompting and evaluation protocol. As shown in Figure 1, on GSM8K, PonderLM-2 is consistently better under majority voting, and remains comparable under best-of-\(N\), at the same inference FLOPs. This suggests that even under matched reasoning-time compute, PonderLM-2 is at least competitive with the standard model.
>
>
>
>
>
>
>
>
> > [W7] Long-Context Evaluation
>
> PonderLM-2 does not reduce the usable context window, since latent thought tokens share the same positional encoding as the corresponding input tokens (Section 3.4). To evaluate this directly, we tested the LLaMA-3-3B continual-pretraining models on RULER at both 4K and 8K context lengths. Empirically, PonderLM-2 consistently outperforms the matched vanilla CPT baseline (74.44 vs. 54.08 at 4K; 61.43 vs. 40.40 at 8K). Moreover, its drop from 4K to 8K (-13.02) is comparable to and slightly smaller than vanilla CPT (-13.68), suggesting that the added latent tokens do not cause a disproportionate degradation in long-context behavior.
>
>
>
> > [Q1] Convergence rate
>
> **Response**: We have measured the convergence of the Jacobi iteration on our LLaMA-3-3B CPT model (https://anonymous.4open.science/status/ponder2_photos-7B41). The RMSE between the iterates H^{(k)} and the sequential fixed point H_seq decreases exponentially. The RMSE drops by over two orders of magnitude within the first 6 iterations and plateaus near machine precision thereafter. This confirms rapid and stable convergence, consistent with the 1.4B analysis.
>
>
>
>
>
> > [Q2] called Jacobi iteration?
>
> Our method shares the same update pattern as classical Jacobi iteration: both solve a fixed-point equation x = Φ(x) by iterating x^{(k+1)} = Φ(x^{(k)}), updating **all components in parallel**. The classical Jacobi method is a special case (linear Φ); we extend it to a nonlinear Transformer operator. The key connection is that all positions/components are updated simultaneously from the previous iterate, rather than being updated one by one using newly updated values, as in Gauss-Seidel.
>
> ---
>
>
> > [Q3] gradients backpropagated
>
>
> Yes, gradients are backpropagated through all K iterations; the hidden states are not detached. This encourages the model to learn hidden states that converge rapidly within a small number of refinement steps.
>
>
>
> > [Q4] more latent thought tokens.
>
> The main-text ablation (Figure 7) uses Pythia-70M (30B tokens) for efficiency. In **Appendix G** (Table 8), we extend to **K=5 on LLaMA-410M** (6B tokens), showing monotonic improvement.
>
>
>
>
>
> > [Q4] "Saturating scaling laws"
>
> We will soften this in the camera-ready version.

---

> > ### Author Rebuttal · Reviewer_KvMB · 2026-04-02
> >
> > I thank the authors for their responses. I maintain my score.

---

> > > ### Author Response · Authors · 2026-04-06
> > >
> > > We sincerely thank the reviewer for confirming that our responses have adequately addressed the concerns. We are grateful for the recognition of PonderLM-2's novelty and extensive experiments. Thank you again for the constructive comments, which has meaningfully improved the paper.

---

### Official Review · Reviewer_XFE5 · 2026-03-16

**Soundness:** 3
**Presentation:** 3
**Significance:** 3
**Originality:** 3
**Overall Recommendation:** 4
**Confidence:** 4

**Summary:**

This article discusses the aspect of scaling computation during the pretraining phase of large language models. Specifically, the authors propose PonderLM-2, a methodology that introduces latent thoughts in a continuous space before generating each discrete token. By utilizing the last hidden state of the current position as the input embedding for the subsequent step, the model learns to refine its predictions. The authors describe their approach as "horizontal scaling", which contrasts with prior "vertical scaling".

To resolve the sequential dependency during training, the authors ingeniously apply Jacobi iteration to parallelize the state updates. The empirical evaluations indicate that PonderLM-2 surpasses standard models with twice the parameters at an identical inference cost, and it outperforms several competitive compute scaling baselines. This research proceeds to assess a relevant issue by demonstrating that the proposed method complements test time scaling strategies like Chain-of-Thought and can effectively enhance existing foundation models through continual pretraining.

**Compliance With Llm Reviewing Policy:**

Affirmed.

**Key Questions For Authors:**

1. Could the authors provide a quantitative comparison of the peak memory usage during training for PonderLM-2 relative to the vanilla model? Understanding how the interleaved sequence and Jacobi iterations impact memory consumption would be highly valuable for practitioners.
2. The analysis in Appendix J is fascinating. Have the authors investigated whether these continuous latent thoughts develop specific attention patterns, or if they act purely as a continuous refinement space? Further probing into what the model attends to during the latent step could offer deeper interpretability.
3. When applying this method to an off-the-shelf model, how smoothly does the model adapt its existing embedding space to accommodate the newly introduced latent states? Adding a learning rate ablation or a discussion on the initial loss spike during continual pretraining would be insightful.

**Limitations:**

See cons.

**Strengths And Weaknesses:**

# Pros
1. The concept of scaling via per token continuous latent thoughts is highly innovative. Applying Jacobi iteration to enable efficient parallel training of this recurrent generative process is a mathematically sound and creative solution that elegantly bridges sequential inference and parallel training.
2. Addressing the saturation of traditional parameter and data scaling laws is a critical endeavor. The approach yields impressive performance improvements across multiple architectures, such as Pythia and LLaMA, on both language modeling and general downstream tasks. The successful continual pretraining of an off-the-shelf 3B model highlights the practical utility of this method for the broader community.
3. The submission provides robust empirical and theoretical validation. The convergence diagnostics of the Jacobi iteration clearly demonstrate its stability and equivalence to sequential inference. The ablation studies, including the analysis of randomized iteration counts and latent thought chain lengths, thoroughly justify the architectural choices.


# Cons
1. The method introduces a training compute multiplier of approximately 8 times compared to the vanilla baseline. While the authors demonstrate superior performance per training FLOP, the paper lacks a detailed discussion on the peak GPU memory footprint required during the parallel training phase. Because the interleaved sequence doubles the context length and requires storing multiple iteration states, a practical analysis of memory overhead would provide a more complete picture of the training requirements.
2. The discussion in Appendix E regarding the randomization of the iteration count is crucial for understanding how the model avoids depth overfitting. Integrating a brief summary of these findings into the main text would enhance the clarity of the training methodology.
3. The evaluations are currently limited to models up to 3 billion parameters. Although the scaling curves are promising, validating the method on a slightly larger scale, such as the 7 billion to 8 billion parameter range, would more conclusively establish its efficacy for modern foundational models.

---

> ### Author Rebuttal · Authors · 2026-03-31
>
> Thank you for recognizing the novelty of scaling computation through per-token continuous latent thoughts, the creativity and soundness of our Jacobi-iteration-based parallel training design, and the strong empirical and theoretical validation of our method. Below, we provide point-by-point responses to your comments.
>
> ---
> > [W1]&[Q1] Memory overhead.
>
> We agree that reporting the peak GPU memory footprint would provide a more complete picture of the training requirements. We have profiled peak GPU memory under the same gradient-checkpointed continual-pretraining setup for both the vanilla baseline and PonderLM-2. The measured overhead is modest: for 3B, vanilla CPT uses 70.9 GB vs. 77.1 GB for PonderLM-2 (1.09×); for 8B, vanilla CPT uses 110.8 GB vs. 124.8 GB (1.13×). Thus, despite the higher training compute, the peak memory increase during training remains limited in practice.
>
> At inference time, the additional memory cost mainly comes from the KV-cache (+1× relative to the 1.4B backbone), which we have analyzed in Appendix I (Table 10). We will add the peak GPU memory comparison to the camera-ready version.
>
> ---
>
>
> > [W2] randomization of the iteration
>
> We agree that this point is important for understanding the training methodology. The main text already briefly mentions the randomization strategy in Section 3.2: *“To prevent overfitting to a fixed number of steps, we randomly sample \(K\) from \(\{2,3\}\) for each training instance.”* In the camera-ready version, we will expand this discussion with the key finding from Appendix E: randomized \(K\) prevents depth overfitting and generalizes smoothly to unseen iteration counts, whereas fixed \(K=2\) exhibits a spike at the first unseen step (\(k=3\)). We will also revise Figure 3 to make the randomization of \(K\) more explicit.
>
> ---
>
>
> > [W3] Limited to 3B, need 7-8B validation.
>
> We agree that larger-scale validation is desirable. While we have not yet been able to run full 8B experiments due to compute constraints, the existing evidence strongly suggests our method scales:
>
> - **From-scratch pretraining** (Figure 10, Appendix B): scaling curves from 405M to 1.4B on both GPT-2 and LLaMA architectures show the performance gap between PonderLM-2 and vanilla **does not narrow** with scale.
> - **Continual pretraining on LLaMA-3-3B** (Section 4.4): PonderLM-2 CPT yields significant improvements over standard CPT across language modeling, downstream tasks, and test-time scaling (CoT, BoN, majority voting).
> - **Memory profiling on 8B**: we have verified that peak memory overhead remains modest at 8B (110.8 GB → 124.8 GB, ~1.13×), confirming no practical memory barrier at this scale.
>
> We plan to include full 8B results in the camera-ready version.
>
> ---
>
>
> > [Q2] Attention analysis.
>
> We have conducted attention analysis on the final forward pass of our LLaMA-3-3B CPT model, measuring where latent thought queries direct their attention across 64 sequences (2048 tokens each, excluding the first two positions to remove attention sink effects).
>
> Latent thoughts develop specific, structured attention patterns. They exhibit a clear layer-wise structure (Figure 2-left): in lower and upper layers, 60–80% of attention mass is directed to original input tokens, while middle layers split attention roughly evenly between original tokens and latent thoughts. Further decomposition (Figure 2-right) reveals that within the attention to original tokens, lower layers broadly gather from earlier positions (context aggregation), while upper layers increasingly focus on the corresponding position's own original token (local refinement). Case studies at the last layer (Figure 3) confirm this pattern: latent queries attend primarily to nearby original and latent tokens.
>
> We will include these attention analysis figures in the camera-ready version.
>
> ---
>
>
>
> > [Q3]  learning rate ablation
>
> We have conducted a **learning rate ablation** for continual pretraining on LLaMA-3-3B, comparing peak LR = 1e-5, LR = 1e-5 (in paper) and 5e-5 (Figure 3). Both settings show the same pattern:
>
> - **Smooth adaptation**: PonderLM-2 exhibits a small, transient loss spike at the very start of CPT as the model accommodates the newly introduced latent states. This spike resolves quickly (within ~0.5B tokens) and **does not recur** during the remainder of training.
> - **PonderLM-2 surpasses vanilla CPT**: after the initial adaptation, PonderLM-2's loss drops below vanilla CPT and remains consistently lower through the end of training (5B tokens). At LR = 1e-5, PonderLM-2 reaches ~2.13 vs. vanilla ~2.14; at LR = 5e-5, PonderLM-2 reaches ~2.14 vs. vanilla ~2.165.
> - **Effect of learning rate**: higher LR (5e-5) leads to faster crossover (~1B tokens) but slightly higher absolute loss; lower LR (1e-5) gives a smoother curve and lower final loss. Both are stable throughout.

---

> > ### Author Rebuttal · Reviewer_XFE5 · 2026-04-06
> >
> > Thank you for the detailed and constructive rebuttal. My main concerns have been adequately addressed.
> >
> > In particular, the newly provided peak memory measurements are very helpful and substantially strengthen the practical case for the method. The reported training memory overhead is modest, at about 1.09x for 3B and 1.13x for 8B under the same setup, which addresses my concern that the higher compute cost might also imply prohibitive memory requirements.
> >
> > The rebuttal resolves my primary concerns without changing my overall recommendation.

---

> > > ### Author Response · Authors · 2026-04-06
> > >
> > > We sincerely thank the reviewer for acknowledging that our responses have adequately addressed the concerns. We also greatly appreciate the reviewer’s valuable comments, which have helped us further improve the paper.

---

### Decision · Program_Chairs · 2026-04-30

**Decision:**

Accept (spotlight)

**Comment:**

This paper proposes PonderLM-2, a pretraining methodology where a language model generates intermediate latent "thoughts" in continuous space before each token prediction, using Jacobi iteration to enable parallel training. In their experiments, the authors found that PonderLM-2 was more compute-efficient, both in terms of parameters and training tokens.

The reviewers unanimously recommend at least weak acceptance. The concerns, including whether the effect on context length was adequately addressed and whether the peak memory requirements were adequately investigated, were fully addressed during rebuttal. This is a clear accept.